# A Catalog of over 5,000 Metagenome-Assembled Microbial Genomes from the Caprinae Gut Microbiota

Xiao-Xuan Zhang,[a,b,h] Qing-Bo Lv,[a,b,h] Qiu-Long Yan,[c] Yue Zhang,[d] Ruo-Chun Guo,[d] Jin-Xin Meng,[a] He Ma,[a] Si-Yuan Qin,[e,f] Qing-He Zhu,[b,h] Chun-Qiu Li,[b,h] Rui Liu,[a] Gang Liu,[a] Sheng-Hui Li,[d,g] Dong-Bo Sun,[b,h] Hong-Bo Ni[a,h]

aCollege of Veterinary Medicine, Qingdao Agricultural University, Qingdao, Shandong Province, China

bHeilongjiang Provincial Key Laboratory of the Prevention and Control of Bovine Diseases, College of Animal Science, Heilongjiang Bayi Agriculture University, Daqing, Heilongjiang Province, China

cDepartment of Microbiology, College of Basic Medical Sciences, Dalian Medical University, Dalian, Liaoning Province, China

dPuensum Genetech Institute, Wuhan, Hubei Province, China

eKey Laboratory of Zoonosis Research, Ministry of Education, College of Veterinary Medicine, Jilin University, Changchun, Jilin Province, China

fCenter for Biological Disaster Prevention and Control, National Forestry and Grassland Administration, Shenyang, Liaoning Province, China

gCollege of Food Science and Nutritional Engineering, China Agricultural University, Beijing, China

hKey Laboratory of Bovine Disease Control in Northeast China, Ministry of Agriculture and Rural affairs of the People's Republic of China, Heilongjiang Provincial Key Laboratory of Prevention and Control of Bovine Diseases, College of Animal Science and Veterinary Medicine, Heilongjiang Bayi Agricultural University, Daqing, China

Xiao-Xuan Zhang, Qing-Bo Lv, and Qiu-Long Yan contributed equally to this work. Author order was determined by contribution to research.

**ABSTRACT** Most microbiome studies regarding the ruminant digestive tract have focused on the rumen microbiota, whereas only a few studies were performed on investigating the gut microbiota of ruminants, which limits our understanding of this important component. Herein, the gut microbiota of 30 Caprinae animals (sheep and goats) from six provinces in China was characterized using ultradeep (>100 Gbp per sample) metagenome shotgun sequencing. An inventory of Caprinae gut microbial species containing 5,046 metagenomic assembly genomes (MAGs) was constructed. Particularly, 2,530 of the genomes belonged to uncultured candidate species. These genomes largely expanded the genomic repository of the current microbes in the Caprinae gut. Several enzymes and biosynthetic gene clusters encoded by these Caprinae gut species were identified. In summary, our study extends the gut microbiota characteristics of Caprinae and provides a basis for future studies on animal production and animal health.

**IMPORTANCE** We constructed a microbiota catalog containing 5,046 MAGs from Caprinae gut from six regions of China. Most of the MAGs do not overlap known databases and appear to be potentially new species. We also characterized the functional spectrum of these MAGs and analyzed the differences between different regions. Our study enriches the understanding of taxonomic, functional, and metabolic diversity of Caprinae gut microbiota. We are confident that the manuscript will be of utmost interest to a wide range of readers and be widely applied in future research.

**KEYWORDS** Caprinae, gut microbiota, metagenome-assembled genomes, microbial function

Sheep and goats are one of the earliest domesticated domestic animals that have been raised for thousands of years (1). Humans can obtain high-value meat, milk, and fur products from ruminants (2). Gastrointestinal microbiome is considered to play a pivotal role in the energy conversion process of ruminants. In addition to influencing host nutrition and metabolism and maintaining homeostasis of lifestyle changes, the gut microbiota can also prevent pathogen invasion and train the immune

Address correspondence to Sheng-Hui Li, lsh2@qq.com, Dong-Bo Sun, dongbosun@126.com, or Hong-Bo Ni, hongboni@126.com.

The authors declare no conflict of interest.

system (3–5). In recent years, due to the affordability of high-throughput sequencing technology and the development of bioinformatic strategies (6), it is now possible to exploit high-quality genomes of rare species in the gut with greater depth of sequencing (7, 8). To date, reference gene catalogs of the gut microbiome have been conducted on the intestinal microorganism of a variety of ruminants. Many metagenomic sequencing data derived from the rumen and gut of cows, buffalo, deer, goats, sheep, and other animals, have been analyzed, and tens of thousands of metagenome-assembled genomes (MAGs) were obtained (9–12). The ruminant microbiota possesses more than 5,000 species of microbes, which is relatively higher than the amount known in the human gut (13). These data indicated a high complexity of the gastrointestinal microbiome of ruminants. Despite a considerable part of metagenomic data focusing on rumen and gut microbiota of Caprinae, a large amount of unknown information is still present in the gut microbiota of Caprinae (9, 14). Understanding and controlling the gut microbiota of Caprinae provides an opportunity to improve production efficiency and reduce feeding costs. It has potential commercial and scientific value and will encourage the use of probiotics in Caprinae farming and even replace the use of antibiotics.

Caprinae animals have species polymorphism, including goats, sheep, antelopes, etc (15). Some of these animals distribute in the wild, and some live with humans. They adapted well to different elevations, geography, climate, and feeding environments. Temperature, vegetation, and water in these habitats significantly affect fiber digestion and nutrient absorption strategies of ruminants (16). A recent study reported differences in gut fermentation patterns between grazing and drylot goats in highland, demonstrating dietary strategies shaping the gut microbiota of goats (14). On the other hand, gut microbiota is largely shaped by a complex interplay of nutrition and genetics, which coevolve with the host genome to adapt to the environment (17). Thus, the host environment results in different gut microbiota structures and functions (18–20). Another example is that wild animals have more complex and varied diets than farm-raised animals, thereby, their guts are prone to contain microbes capable of digesting different substrates (21–23). In conclusion, the collection of cross-regional samples will contribute to the discovery of more unknown species of microbiota in the Caprinae gut.

Here, the whole-metagenomic sequencing was employed to investigate the microbial communities in the feces of Caprinae breeds (including *Capra hircus*, *Ovis aries*, *Pantholops hodgsonii*, and *Procapra picticaudata*) from six provinces of China (including Anhui, Shanxi, Shandong, Jilin, Guangxi, and Tibet). A genomic catalog of gut microbiota containing 5,046 MAGs was constructed. This catalog includes more than two thousand novel bacterial and archaea species. In addition, we also analyzed the functional specificity of these species in each of the different regions. These discoveries expand our understanding of the gut microbiota of Caprinae and provide new insights into the relationship between the gut microbiota of Caprinae and the host growth environment.

## RESULTS

**Metagenome-assembled genomes from the Caprinae gut.** To characterize the microbiota in the Caprinae gut, a total of 3.3 Tb of high-quality Illumina sequencing data (110.1 ± 8.3 Gb per sample; see Table S1 in the supplemental material) were produced from 30 Caprinae fecal samples collected from six distinct regions across China (Fig. 1A). On average, 2.0% of sequences were originated from the hosts and removed for analysis. We performed metagenomic assembly and binning for each sample, and a total of 22,882 raw bins were generated with a minimum length of 200 kb. Compatible bins with approached sequencing coverage (±10%) and G+C content (±2%) and identical taxonomic assignment were merged. A total of 5,046 MAGs were yielded with estimated completeness ≥70%, contamination <5%, and quality score (defined as completeness −5 × contamination) (24) above 55 (Table S2). All the MAGs met or exceeded the medium-quality "minimum information about a metagenome-assembled

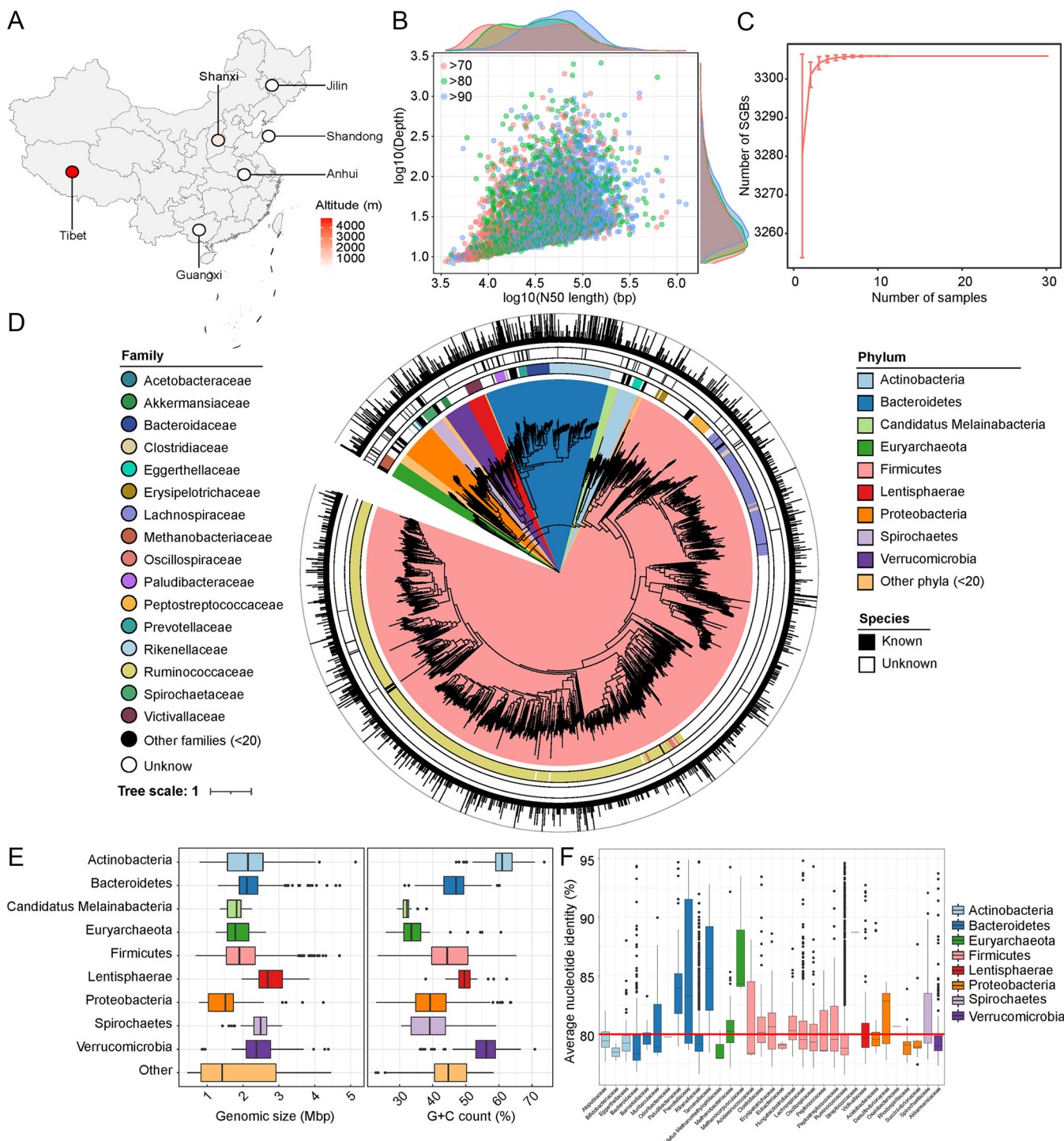

**FIG 1** Statistics of samples and MAGs. (A) The geographical diagram of the sampling sites, with the gradient in red representing the altitude of the sampling sites. (B) The relationship between sample sequencing depth and $N_{50}$ length of MAGs. Colored points represent the MAGs with different completeness: red, completeness >70%; green, completeness >80%; and blue, completeness >90%. (C) Rarefaction curve analysis shows the function of the number of samples and the number of observed SGBs. (D) A phylogenetic tree of 3,306 SGBs, with color filling on branches representing phylum-level classification. The first outer ring represents the family-level classification of the branches, and the second ring represents whether the branches have been classified into the species level. The outermost bar chart shows the genome size of each SGB (if the genome is larger than 5 M, it is drawn as 5 M). (E) Boxplot shows the genome sizes and G+C contents of the top 10 most abundant phyla. F. Boxplot shows the pairwise average nucleotide identity (ANI) within each family.

genome" (MIMAG) standard of bacteria and archaea (25). Of these, 1,933 MAGs reached the high-quality standard (completeness ≥90% and contamination <5%), and 3,113 were identified to be medium quality. The average completeness and contamination of MAGs were 86.3% and 1.4%, respectively. Notably, 91.6% of MAGs had an assembly $N_{50}$ length >10 kb (Fig. 1B), highlighting the high assembly quality of these MAGs.

The MAGs were grouped into 3,306 species-level genome bins (SGBs) based on the average nucleotide identity (ANI) threshold of 95%, which was taken as the species boundary of the prokaryotic genome (26). These SGBs were a good representation of the species in gut (Fig. 1C). A phylogenetic tree of the SGBs was shown in Fig. 1D. Taxonomic classification of SGBs was performed using the Genome Taxonomy Database (GTDB R89) and manual classification adjustment, and 3,263 (98.7%) SGBs were assigned to bacterial phyla, including 2,505 *Firmicutes*, 349 *Bacteroidetes*, 104 *Proteobacteria*, 71 *Verrucomicrobia*, 57 *Actinobacteria*, 55 *Lentisphaerae*, 38 *Spirochaetes*, and 84 members belonging to the other 10 phyla (Table S3). The remaining 43 (1.3%) SGBs belonged to *Euryarchaeota*, an archaeal phylum that is widely present in human and mammal gut microbiotas (27). Additionally, 3,277 SGBs were classified into the order level; *Clostridiales* ($n = 1,824$) and *Bacteroidales* ($n = 348$) were the dominant orders. At the family level, 95.6% (3,161/3,306) SGBs could robustly be assigned into bacterial or archaeal families. The dominant families were *Ruminococcaceae* ($n = 1,314$), *Lachnospiraceae* ($n = 320$), and *Rikenellaceae* ($n = 163$). Only 14.3% SGBs could be classified into the genus level, including *Ruminococcus* ($n = 86$), *Alistipes* ($n = 83$), *Treponema* ($n = 34$), *Methanobrevibacter* ($n = 23$), and *Akkermansia* ($n = 21$). Most of the archaeal genomes were predicted to be members of *Methanobrevibacter* ($n = 23$) and *Methanosphaera* ($n = 12$). The coverage of SGBs in each sample was shown in Table S4. Using a cutoff 1% coverage, 47.2% (1,561) SGBs were present in 90% of samples, and all samples shared 795 SGBs. Most of these core SGBs belonged to members of *Ruminococcaceae* ($n = 383$), *Lachnospiraceae* ($n = 140$), and CAG-74 ($n = 69$). In addition, only 20 SGBs were uniquely presented in a single sample, mostly belonging to members of *Firmicutes* and *Proteobacteria*.

Of the 3,306 SGBs, only 88 showed >95% ANI with available sequenced genomes in National Center for Biotechnology Information (NCBI) (Fig. 1D) (28), and the others were identified to be novel species in the Caprinae gut microbiota. The 88 known species included *Fibrobacter succinogenes*, *Bacteroidales bacterium*, *Alistipes shahii*, *Ruminococcus bromii*, *Methanobrevibacter smithii*, and *Prevotella ruminicola*, and most of these species were known members in human or other animal gastrointestinal tracts (29–33).

Focusing on the genome characteristics of the microbial species in the Caprinae gut, we first revealed a diverse distribution of the genome size and G+C content of the 3,306 SGBs (Fig. 1E; Fig. S1). Members of *Actinobacteria* had the highest G+C content (average 60.5%) versus species of other phyla, while members of *Planctomycetes* had the highest genome size (average 4.4 Mb). In addition, we found that the members of *Proteobacteria* had the smallest genome size (average 1.7 Mb) in the Caprinae gut. In fact, unlike the domination of *Enterobacteriaceae* in the human gut, the members of *Proteobacteria* in Caprinae gut were composed of *Alphaproteobacteria* (72.1%, 75/104), *Desulfovibrionaceae* (8.7%, 9/104), and *Succinivibrionaceae* (7.7%, 8/104). All these members represented relatively small genomes (Fig. S2). Next, we noted that 11 of 32 families had an average interspecies ANI less than 80% (Fig. 1F), thus revealing a high nucleotide diversity of the Caprinae gut species at the low taxonomic level. This finding could be at least attributed to the following reasons: (i) the endogenetic high genetic diversity for some taxa and (ii) the undiscovery of a vast number of species in the Caprinae gut. Despite that, members of *Prevotellaceae*, *Paludibacteraceae*, *Tannerellaceae*, and *Methanocorpusculaceae* had relatively high interspecies diversity compared with other families.

**Expansion of the mammal-associated microbial tree of life.** To extend the content of Caprinae gut microbial genomes into the comprehensive mammal-associated microbial tree of life, the 3,306 Caprinae gut SGBs were compared with four large-scale available data sets: (i) the genome catalog containing 4,941 MAGs of rumen microbial

in ruminant cattle (RUG) (11); (ii) the genome catalog containing 4,644 nonredundant species of the human gut microbiome (Unified Human Gastrointestinal Genome, UHGG) (34); (iii) the genome catalog containing 2,985 MAGs of nonhuman primate (NHP) gut microbiome (35); and (iv) the genome catalog containing 8,745 species-level genomes of the nonredundant ruminant gastrointestinal tract (RGIG) (9). Each data set was classified by using the GTDB database (36, 37). These catalogs were integrated, and then, all genomic data was dereplicated using dRep (38). At 95% ANI, the integrated catalog was reduced to 16,036 clusters (Table S5). The RUG, UHGG, NHP, and RGIG data sets included 2,162; 4,178; 1,313; and 6,417 nonredundant species, respectively, in an aforementioned species-level clustering manner.

*Bacteroidetes*, *Firmicutes*, *Actinobacteria*, and *Proteobacteria* were the dominant species in nearly all mammalian gut (Table S6). However, the Caprinae gut had a significantly higher proportion of *Firmicutes* members and lower proportions of *Bacteroidetes*, *Actinobacteria*, and Proteobacteria members compared with the other four habitats (Fig. 2A). The proportions of the rare phyla also differed between Caprinae gut and other animal guts; members of *Verrucomicrobia* and *Saccharibacteria* were primarily found in the Caprinae gut, whereas the *Euryarchaeota*, *Spirochaetes*, *Candidatus Melainabacteria*, and *Kiritimatiellaeota* members were depleted in the Caprinae gut. In addition, *Fusobacteria* was not found in the Caprinae gut. The family clades were also significantly differed in proportions between Caprinae gut and the gut of the other four animals. A striking example was that *Ruminococcaceae*, representing nearly 40% (1,314/3,306) of the species in the Caprinae gut, was significantly higher in the gut of other animals (RUG: 453/2162, 21%; UHGG: 368/4178, 9%; NHP: 72/1313, 5%; RGIG: 278/6417, 4%). Another example was *Rikenellaceae*, which represented 163 (~5%) species in Caprinae gut and 187 species in RGIG (~3%), but merely 80 species in the gut of other animals (Fig. S3). Moreover, ANI comparison at the species level showed a consistent result. It was observed that the species in the Caprinae gut had low overlap with other catalogs (Fig. 2B). Our data set shared with two ruminant data sets (RUG and RGIG) was 55 and 659 species, respectively. Among them, the overlap with RGIG may be because partial genomes in the data set came from the gut of ruminants. This finding indicated that at least 76.5% (2,530/3,306) SGBs in the data set of this study were novel gut species that had not been identified yet. These Caprinae-gut-enriched or -specific taxa might be linked to the genetic, geography, and lifestyle specificity of Caprinae.

A huge number of Caprinae-specific taxa were rarely described in other habitats. Here, we found some interesting phenomena of these underrepresented taxa. Strikingly, the Caprinae gut had 71 *Verrucomicrobia* species, 21 of which belonged to *Akkermansia*, the only discovered genus (containing only two species, *A. muciniphila*, and *A. glycaniphila*) of *Verrucomicrobia* in the human and animal guts (39). We found that the 21 Caprinae *Akkermansia* SGBs present a distant evolutionary distance from known *Akkermansia* members (Fig. 2C). The high diversity of *Akkermansia* communities in the gut might suggest that the Caprinae gut acted as an evolutionary repertory of multiple *Akkermansia*/*Verrucomicrobia* species. In this situation, human and other animals' *Akkermansia* species might evolve from Caprinae gut homologous stains. Therefore, a bold hypothesis is that *Akkermansia* is captured and retained by humans as an exotic bacterium during human contact with animals, which may explain the extremely low diversity of *Akkermansia* species in the human gut. Another example is a majority of 34 *Treponema* species observed in the Caprinae gut were crossly presented in the phylogenetic tree with *T. succinifaciens* and *T. brennaborense* (Fig. 2C). Considering that the adjacent species of *T. succinifaciens* and *T. brennaborense* were rarely studied, we proposed that the Caprinae gut might be also an evolutionary repertory of these species. Furthermore, the Caprinae gut also likely acted as a repertory of *Eggerthellaceae* (containing 26 species; Fig. 2C), *Euryarchaeota* (containing 43 species; Fig. 2c), etc. These findings and future descriptions of these underrepresented taxa would greatly expand the knowledge of the mammal-associated microbial tree of life.

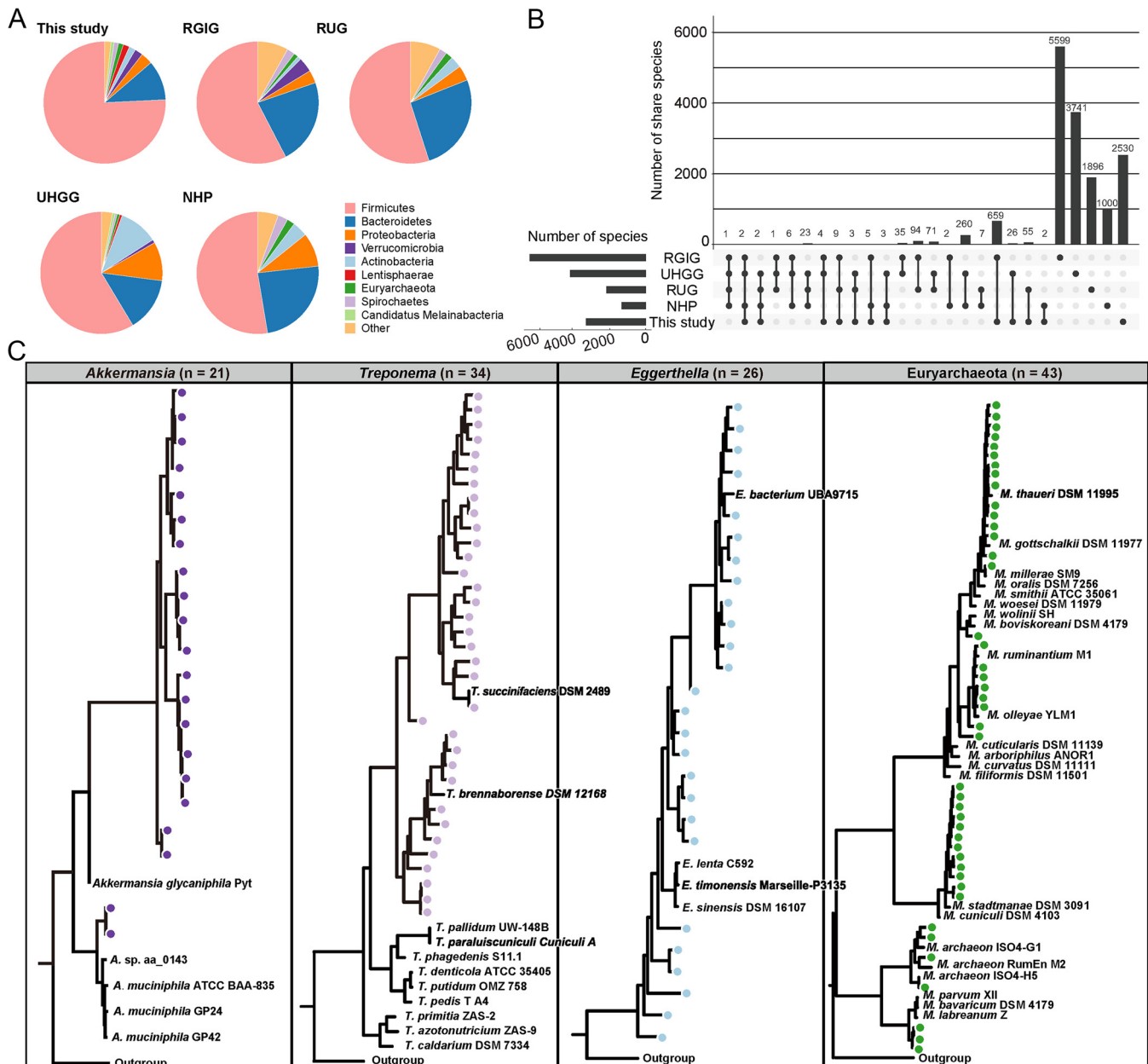

**FIG 2** Comparison of five microbial genome catalogs and phylogenetics of the Caprinae-specific taxa. (A and B) show the distribution and sharing of species among 5 microbial genome catalogs. (C) The phylogenetic tree of several Caprinae-specific branches, including *Akkermansia*, *Treponema*, *Eggerthellales*, and *Euryarchaeota*. Nodes represent the MAGs from the Caprinae gut, and the other homologous strains are downloaded from the NCBI RefSeq genomes.

**Functions of Caprinae gut microbiota genomes.** To characterize the functions of the Caprinae gut microbiota, we performed a comprehensive functional annotation of the 5,046 MAGs by using Kyoto Encyclopedia of Genes and Genomes (KEGG) (40), evolutionary genealogy of genes: Nonsupervised Orthologous Groups (eggNOG) (41), biosynthetic gene clusters (BGC), and CAZy (carbohydrate-active enzyme database) (42) (Table S7). On average, 60.7% of the genes in each genome were annotated with the KEGG database, covering a total of 7,973 KEGG orthologs (KOs) and 342 modules. The proportion of annotated genes was relatively higher in *Actinobacteri*a (average, 66%) and *Proteobacteria* (average, 69%), but lower in *Verrucomicrobia* (average, 54%) (Fig. 3A). Some rare phyla might lack annotations, such as *Planctomycetes* (45.9%), *Candidatus Eremiobacteraeota* (47.3%), and *Tenericutes* (51.4%) (Fig. S4). Principal coordinates analysis (PCoA) showed that the functional profiles of MAGs were largely determined

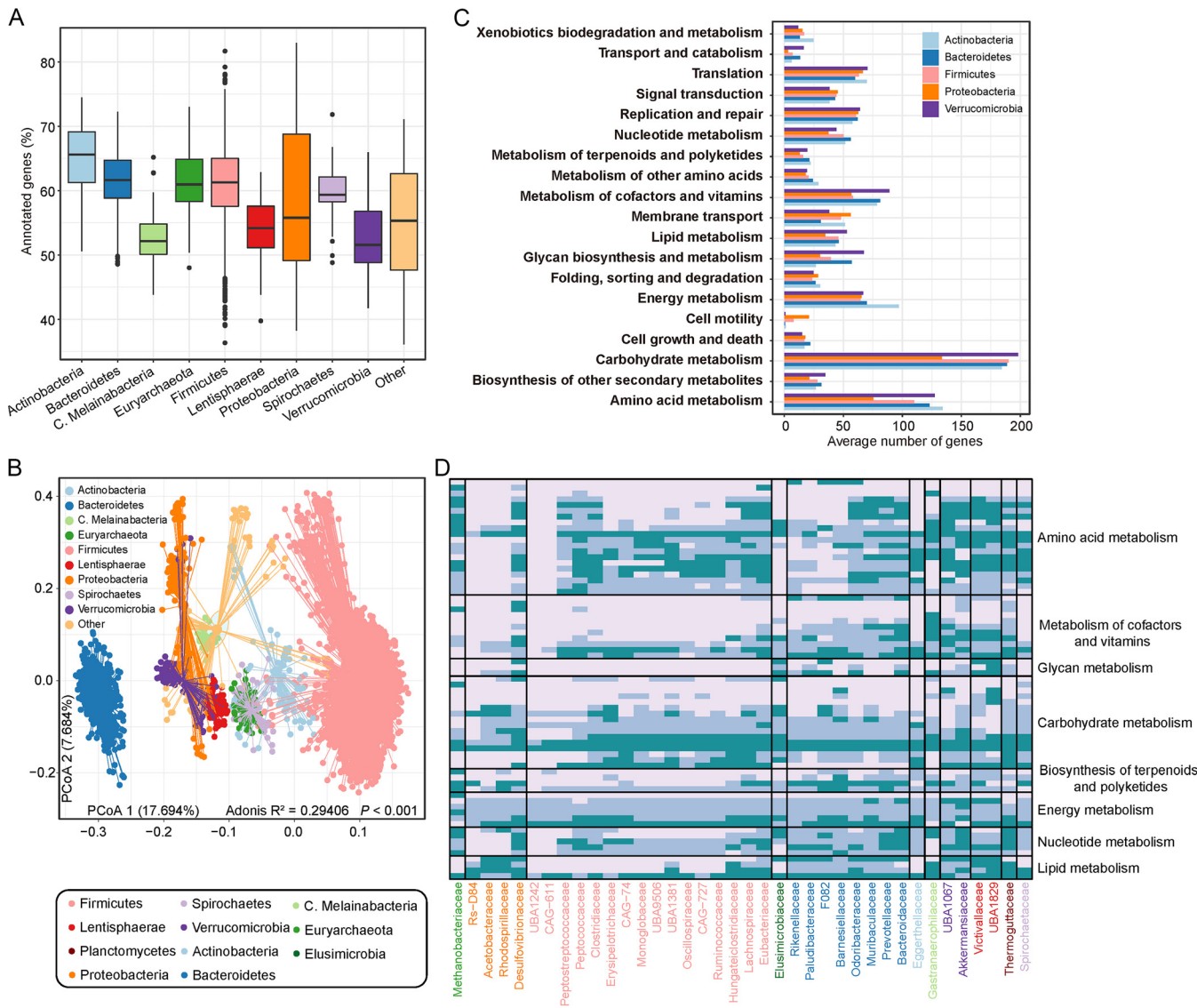

**FIG 3** KEGG profiles of the Caprinae gut MAGs. (A) Boxplot shows the proportions of KEGG-annotated genes of the MAGs at the phylum level. (B) Principal coordinates analysis (PCoA) of Bray-Curtis distance of the KO profiles of all MAGs. Different phyla are shown at the first and second principal coordinates (PCoA 1 and PCoA 2), and the ratio of variance contributed by two principal coordinates is shown. Permutational multivariate analysis of variance (PERMANOVA) shows the effect size ($R^2$) of taxonomy that contributes to the variance of the KEGG functions. (C) Average number of genes involved in the KEGG pathway of five major phyla. (D) Heatmap shows the integrality of metabolic modules (dark to light representing "complete," "partial," and "absent") of MAGs at the family level.

by their phylum-level stratification (*adonis* $R^2 = 0.28$, $P < 0.001$; Fig. 3B). Also, the function of MAGs was still dependent on their phylogenetic position at the family level (*adonis* $R_2 > 0.34$, $P < 0.001$ for all levels; Fig. S5). A comparison of the KEGG pathways (level B) revealed a remarkable diverseness of functions for the MAGs (Fig. 3C); for instance, (i) *Firmicute*s and *Proteobacteria* had a higher proportion of genes involved in cell motility compared with other phyla; (ii) *Verrucomicrobia* had the highest number of genes involved in glycan biosynthesis and metabolism and lipid metabolism; (iii) *Actinobacteria* had the highest number of genes involved in xenobiotics biodegradation and metabolism and energy metabolism. In particular, the *Proteobacteria* members had a remarkable reduction of genes that were involved in metabolism pathways.

Next, we analyzed the KEGG level C pathways and focused on metabolism-related functions. These bacteria in the gut have wide functions, including purine metabolism, amino/nucleotide sugar metabolism, and pyrimidine metabolism (Table S8). In

addition, the functional profiles of the five major phyla were compared based on the characterized functions encoded by more than 90% of the members of each phylum (Fig. S6; Table S9). Notably, a lot of *Actinobacteria* members included genes that related to sulfur metabolism in actinomycetes (ko00920), which had not been reported in other ruminant data. In addition, the *Actinobacteria* bacteria encoded valine in large numbers (Table S9), leucine and isoleucine degradation (ko00280), glutathione metabolism (ko00480), ubiquinone and other terpenoid-quinone biosynthesis (ko00130). Compared with other phyla, CD molecules (ko04090) (fold change = 87) and protein digestion and absorption (ko04974) (fold change = 36) were more frequently encoded by members of *Bacteroidetes*. The proteins associated with bacterial motility (ko02035) and lipopolysaccharide biosynthesis (ko00540) were encoded by the members of *Proteobacteria*.

At the KEGG module level, 69 metabolism-related modules had differences in integrity in different phyla or families (Fig. 3D). Several modules, including M00021 (cysteine biosynthesis), M00005 (PRPP biosynthesis), M00307 (pyruvate oxidation), M00579 (phosphate acetyltransferase-acetate kinase pathway), and M00086 (beta-oxidation), were with the higher average integrity among all families. These modules of bacteria completed the core functions of life activities that related to amino acid metabolism, carbohydrate metabolism, energy metabolism, metabolism of cofactors, and lipid metabolism. *Bacteroidaceae* had four complete nucleotide metabolism modules that were involved in purine and pyrimidine metabolism, whereas these modules were completely absent in the CAG-611 and UBA1242 families and presented at lower levels in other families (Fig. 3D). These results suggested the functional uniqueness of *Bacteroidaceae* in the aspect of nucleotide metabolism. The integrity of some functional modules showed a certain regularity at the family level. *Firmicutes* (almost *Clostridiaceae*) lacked many lipopolysaccharide and lipid metabolism-related modules. However, Clostridiaceae, *Lachnospiraceae*, and *Ruminococcaceae* were responsible for encoding carbohydrate-degrading enzymes in the intestine and were highly concerned producers of intestinal short chain fatty acid (SCFA). It was observed that these bacteria had high integrity in amino acid metabolism and carbohydrate metabolism-related modules, such as M00854 (glycogen biosynthesis), M00844 (arginine biosynthesis), M00010 (citrate cycle, first carbon oxidation), etc. In addition, consistent with the previously observed results, the integrity of *Proteobacteria* in metabolic modules, except for lipid metabolism, was lower than that of other families. This might be related to their lifestyle, some *Proteobacteria* can grow at a very low nutrient level, suggesting metabolic function might not be greatly required for sustaining themselves.

A total of 5,586 secondary metabolite biosynthesis gene clusters (BGCs) were identified from the Caprinae gut MAGs (Table S10). These BGCs were grouped into 5 major categories and 32 product types. The main products were sactipeptide (34.9% BGCs of a known class), betalactone (20.7%), and nonribosomal peptide synthetase (NRPS) (12.3%). In addition, arylpolyene (8.1%), NRPS-like (7.5%), and terpene (4.7%) were also present in small amounts (Fig. S7). Notably, *Ruminococcaceae* specifically encodes sactipeptides, compounds with antimicrobial activities belonging to ribosomally synthesized and posttranslationally modified peptides (RiPPs). Balty et al. (43) reported ruminococcin C1 and C2, a class of sactipeptides in the *Ruminococcus gnavus* species, were against Gram-positive bacteria *Clostridium perfringens* and *Bacillus subtilis*, and induced a lag phase for Gram-negative bacteria *Escherichia coli*. In addition, BGCs encoding aryl polyene (44) molecules in several Bacteroidetes (244 BGCs), *Lentisphaerae* (65 BGCs), and *Verrucomicrobia* (50 BGCs) genomes were identified. This is a highly unexplored (45) class of small molecule compounds that may act as a protective agent against oxidative stress.

We finally investigated the distribution of antibiotic-resistant genes in the gut microbiota of Caprinae. This analysis identified 69 antibiotic-resistant genes from 46 microbial strains (Table S11), including *vanG*, *vanR-O*, *aph* (2′ )-*IIa*, *acrE*, etc. These resistance genes were mainly carried by *Proteobacteria*, *Firmicutes*, and *Bacteroidetes* members, and a small amount was also carried by *Verrucomicrobia* and *Actinobacteria*. We also annotated the microbial virulence factors by searching in the VFDB database

(46) and identified 302 virulence genes from 17 MAGs (Table S12). Most of the MAGs containing virulence genes belonged to *Proteobacteria*, thus suggesting that the phyla might contain more potentially pathogenic bacteria.

**Identification and description of carbohydrate-active enzymes (CAZymes).** A total of 960,735 CAZyme-encoding genes (representing 344 CAZy families; Table S13) were obtained from 5,046 Caprinae gut MAGs. On average, 10.7% of genes in each genome can be assigned into a CAZyme-encoding gene. These genes included 399,706 glycoside hydrolases (GHs), 357,647 glycosyltransferases (GTs), 57,299 carbohydrate esterases (CEs), 126,100 proteins with carbohydrate-binding modules (CBMs), 10,290 polysaccharide lyases (PLs), and 9,693 auxiliary activities (AA). Among the GTs, the subfamilies GT2 and GT4 were two major members, the number of involved genes was far greater than that in other CAZymes. CBM50 had the highest content of CBM. Members of CBM50 can bind to *N*-acetylglucosamine, a structure in carbohydrates, e.g., bacterial peptidoglycans and chitin (47).

The phylogenetic tree showed the CAZyme-encoding genes were presented in major families (Fig. 4; Table S13). *Firmicutes* ($n = 652,010$) and *Bacteroidetes* ($n = 181,114$) together contributed to the largest number of CAZyme proteins in our data set (Table S14), and the proteome of *Bacteroidetes* and *Firmicutes* had 11.8% and 11.6% CAZyme activity, respectively (Table S15). Interestingly, there were only a few studies showing the role of *Lentisphaerae* in the mammal gut habitats previously; however, *Lentisphaerae* devoted a higher proportion of its genes to carbohydrate metabolism in our data set (12.1%). *Fibrobacteres* has been identified as an important cellulose-degrading bacterium in the rumen, which mainly includes GH5 and GH9 family genes and has a high-cellulolytic activity (48). However, the abundance of *Fibrobacteres* in the gut of Caprinae seemed to be quite small, and only 6 MAGs in our data set belonged to this phylum. At the family level, *Ruminococcaceae* and *Lachnospiraceae* provided the most CAZyme proteins, and followed by *Rikenellaceae* and *Bacteroidaceae*. We also focused on the degradation of plant polysaccharides by these microbiotas. Of the six common polysaccharides, most bacteria were involved in the metabolic degradation of pectin, xylan, and mannan; however, xyloglucan and starch were not degraded by these bacteria (Fig. S8).

**Geographical environment adaptation in microbial composition and function.** To analyze the gut microbiota characteristics of samples from different regions, the bacterial community composition of these regions was compared. PCoA analysis based on Bray-Curtis distance showed that there were significant separations among samples from different provinces, with a significant correlation between regional factors in the top two principal coordinates (PC1 and PC2) (Fig. 5A). Permutation multiple variance analysis (PERMANOVA) showed that geographic factors were responsible for 33.2% of variations of the gut microbiota, while species, altitude, and sex were responsible for 16.1%, 8.8%, and 8.7% of variations, respectively. These results suggested that geographical factors were the main reasons for gut microbiota changes. However, we did not find any significant difference in the diversity of samples from different regions, and the species richness and evenness were roughly the same in different regions (Fig. 5B). At the family level, *Ruminococcaceae* and *Rikenellaceae* were the dominant ones in almost all the samples. In addition, the samples from Tibet had more *Ruminococcaceae* bacteria and fewer *Prevotellaceae* and *Rikenellaceae* (Fig. 5C), and changes in the abundance of these bacteria were associated with plateau adaptation (49). Next, the prevalence of species between each geographical region were compared. A species must have a relative abundance of 0.01% in at least 50% of samples from a region to be considered universal in samples from that region. A total of 260 species were found to be shared among samples from the six regions (Fig. 5D; Table S16), indicating the existence of a common core gut microbiota among these Caprinae animals. In addition to *Firmicutes* and *Bacteroidetes* members, three *Euryarchaeota* members (SGBs0080, SGBs0412, and SGBs0530) and one *Proteobacteria* member (SGBs1934) were also found in this core group. Consistent with the diversity analysis, the endemic species appeared mostly in Tibet ($n = 182$) and Guangxi province ($n = 182$) (Fig. 5D).

To further discuss the gut microbiota characteristics in Tibetan areas, all Tibetan

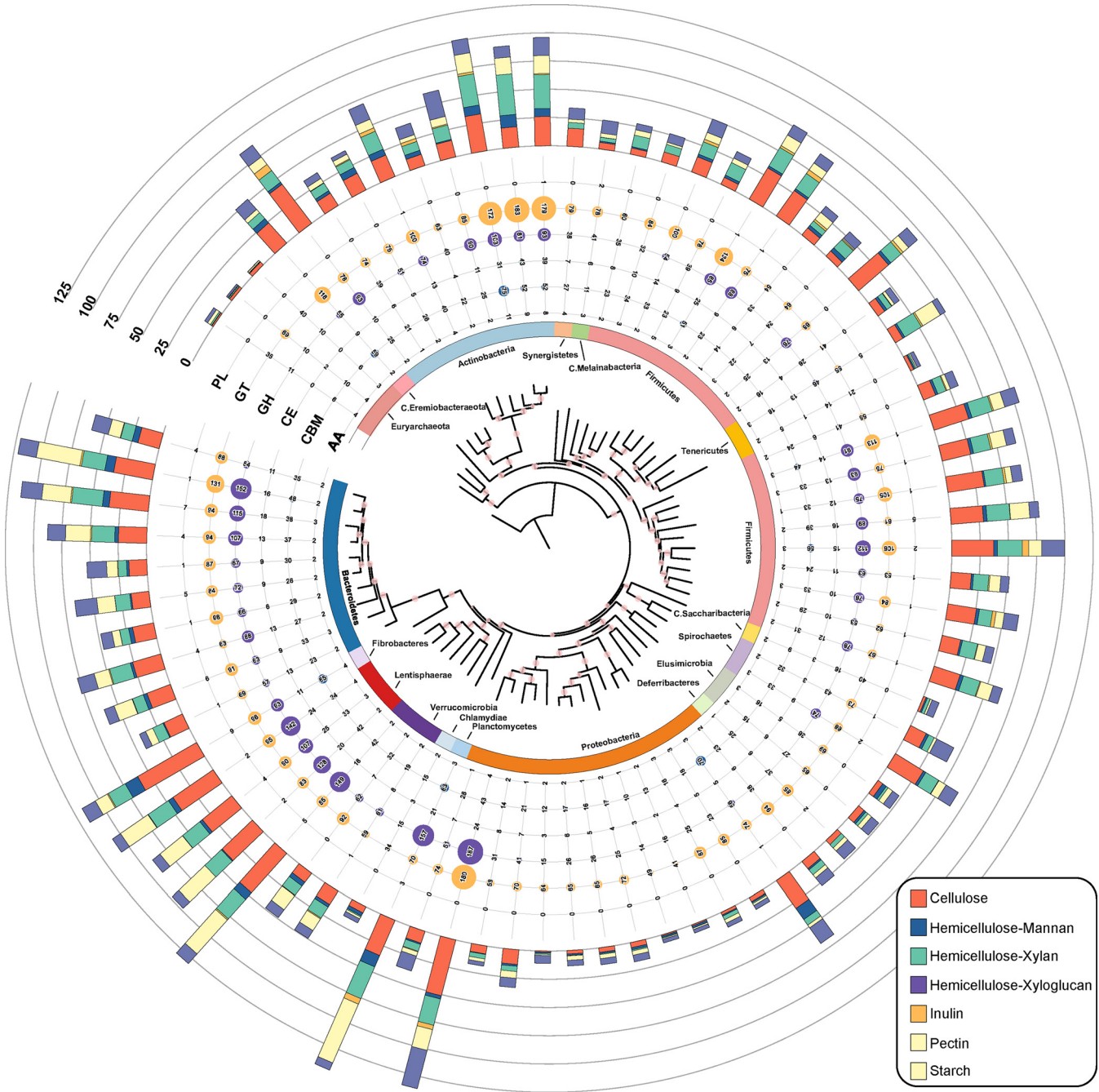

**FIG 4** Overview of CAZymes and BCGs of the Caprinae gut MAGs. Each clade represents a family, the outer ring represents phylum-level classification, the dot shows the median number of CAZymes genes carried by MAGs in each family, and the outermost stacking bar chart shows the number of genes involved in the metabolism of six substrates in this family.

samples ($n$ = 5) were compared with non-Tibetan samples ($n$ = 25). PERMANOVA showed that the Tibetan environment was responsible for 8.8% of the total microbial variation ($P$ < 0.05). Our own data set was employed to further characterize the microbiota structure of Caprinae from Tibetan and non-Tibetan areas. A total of 186 SGBs were observed to be abundant in the Tibetan group and 81 were abundant in the non-Tibetan group. Animals in the Tibetan area were enriched with *Firmicutes* (mainly *Ruminococcaceae*), *Actinobacteria* (mainly *Eggerthellaceae*), and *Euryarchaeota*, while *Bacteroidetes* were relatively less (Fig. 6A). This resulted in a higher F/B ratio in the gut of animals in the Tibetan group, and the same microbiota pattern also existed in the

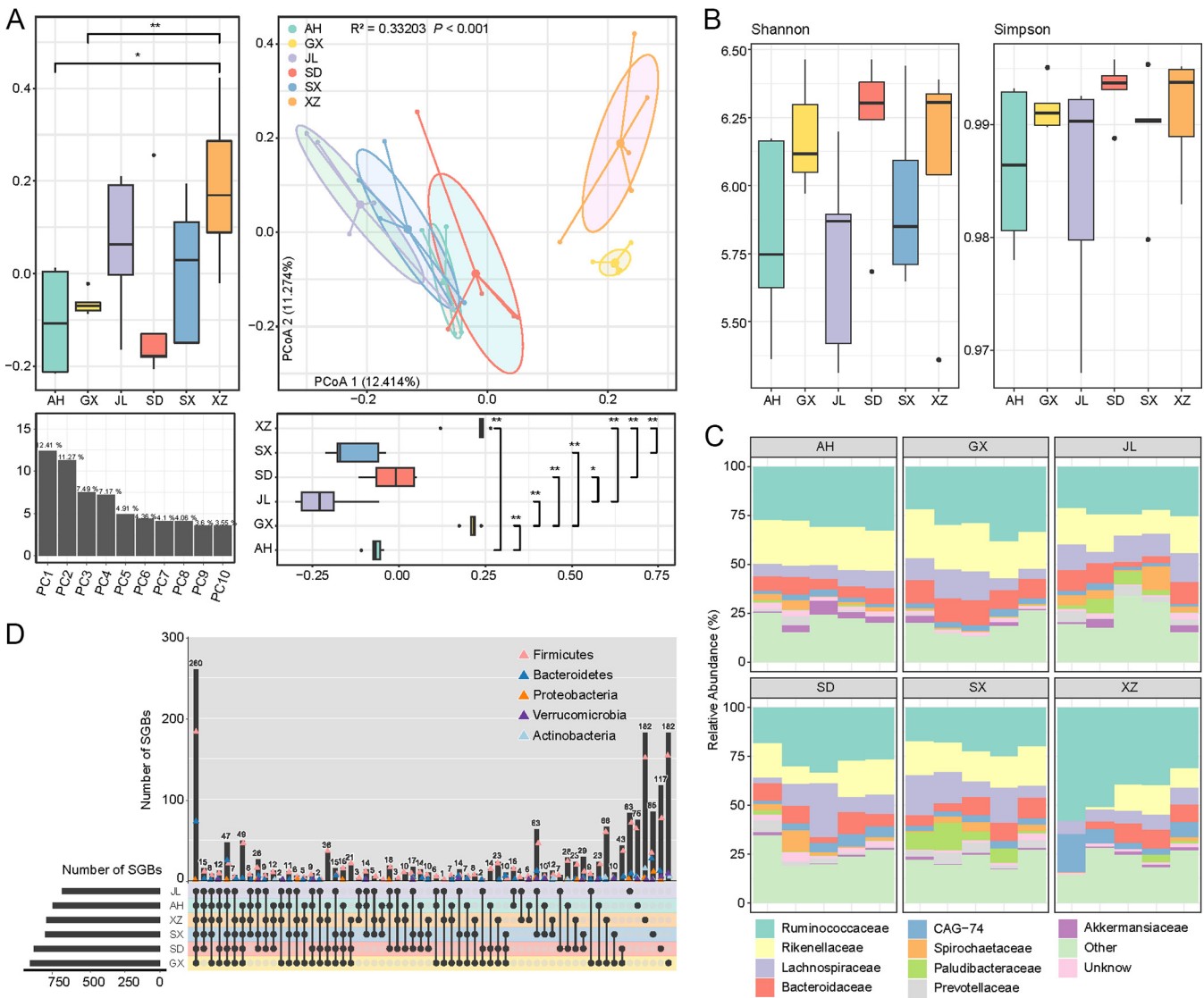

**FIG 5** Comparison of Caprinae gut microbiome among different geographies. (A) PCoA of Bray-Curtis distance on gut microbial species from six provinces, and ellipsoid represents the 95% confidence interval around each group. The boxplots on the left and below show the explainability of the samples in each province at the first and second principal coordinates, and the results of the significance test are showed. *, permutated $P < 0.05$; **, permutated $P < 0.01$. The bar chart in the lower-left corner shows the degree of interpretation of the first 10 coordinates. (B) The $\alpha$-diversity of each group of samples is demonstrated by Shannon and Simpson diversity indexes. (C) Accumulation bar chart shows the family level composition of each grouping sample. (D) The UpSet diagram shows the shared species of each group. The trigonometric symbol represents the number of five important phyla in the shared bacteria, and the higher position, the higher the number.

gut of yaks and Tibetan antelopes (49), which might be associated with the ability to adapt to the extreme environment of the plateau.

In addition, the functional characteristics of gut microbiota in Tibetan and non-Tibetan areas were compared. Compared with non-Tibetan samples, Tibetan animals had more metabolism-related modules, especially nucleotide metabolism and amino acid metabolism. The modules degrading purines, tyrosine, lysine, histidine, and homo-protocatechuate were enriched in Tibetan animals, whereas those for biosynthesis ribo-flavin, histidine, tryptophan, and triacylglycerol were enriched in non-Tibetan animals. Moreover, bacteria in Tibetan animals to participate in the synthesis of cobalamin, cholesterol, fatty acid, ergocalciferol, bile acid, menaquinone, glycosaminoglycan, GABA, heme, li-poic acid, etc than that in non-Tibetan animals (Fig. 6B). Notably, M00356 and M00567 were found to be enriched in the Tibetan group. These modules were key for methanogenesis (Fig. 6B). Enzymes in these modules are widely encoded by methanogen, such as

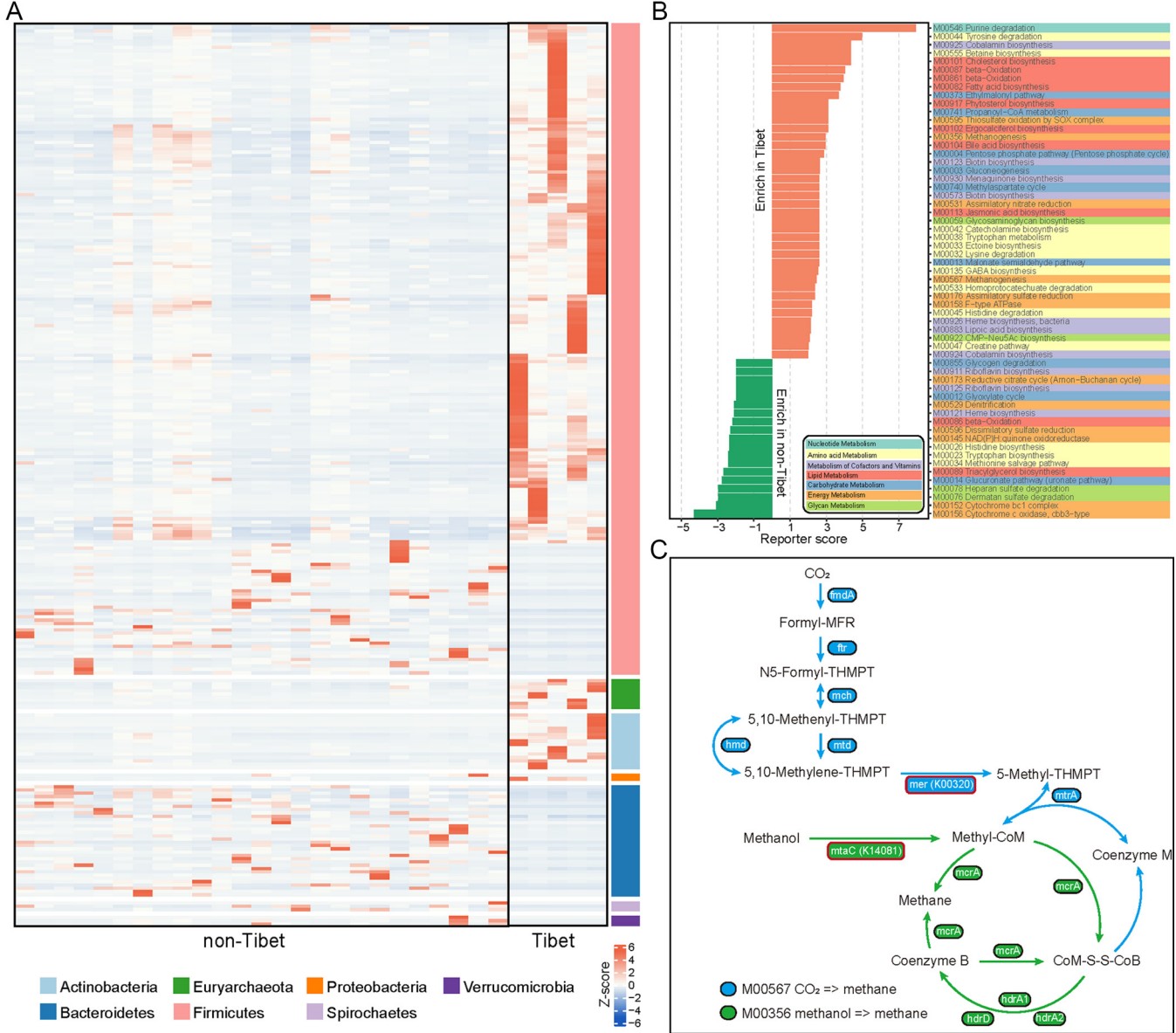

**FIG 6** Caprinae gut microbiome composition and functional comparison of Tibetan and non-Tibetan groups. (A) Heatmap showing the difference in distribution at phylum level between the two groups. (B) Report score evaluated the enrichment of modules between the two groups. (C) Metabolic processes of two modules M00567 and M00356 are shown, with the red border indicating the enzymes enriched in the Tibet group.

*Methanosphaera* and *Methanobrevibacter* (50), and they were significantly abundant in the Tibetan group (Table S18). The genes encoding K00320 and K14081 were significantly increased in the Tibetan group, mainly affecting the synthesis of 5,10-Methylene-THMPT to 5-Methyl-THMPT and the synthetic process of methanol to Methyl-CoM (Fig. 6C).

## DISCUSSION

In this study, the gut microbiota of Caprinae from 6 provinces in China was deeply analyzed based on metagenomic sequencing and the construct of a microbial genome catalog containing thousands of unknown species of gut microbiota. These MAGs had integrity greater than 70% and contamination rates less than 5% and could be classified as medium-high quality sketch genomes based on Bowers et al. (25) standards. The Caprinae gut microbial genome catalog largely complemented and extended the existing reference data sets of ruminants (9, 11, 12), thus providing clues for future

acquisition of uncultured microbiota and a full understanding of the complex ruminant gut environments.

The number of clean reads per sample averaged 110.1 Gbp, which was a large sequencing throughput compared to previous studies. Contrary to our expectations, the increase in sequencing volume did not significantly increase the utilization of reads. In brief, the low reads utilization rate suggested that there were still many species that had not been sequenced and assembled. Even so, the data set further expanded the number of known Caprinae gut species by adding more than 2,000 new species representing four important Caprinae species. In addition, species composition analysis also revealed the unique gut microbiota structure of Caprinae in this study, such as the high-abundance *Ruminococcaceae*, which is significantly different from the core gut microbiota in other animals (9, 11, 24, 34). Our results suggested that the gut microbiota of these ruminants is more complex and diverse than that of humans or other animals. The complex characteristics of the Caprinae gut environment may be closely related to dietary characteristics. The food sources of Caprinae in the wild can be complex. In nonrearing conditions, ruminants, due to their physiological characteristics of rapid feeding, ingest many substances that are mixed with plants, including remains of animals, insects, fungi, and soil. Therefore, future research needs to include more environmental information, such as investigating the food sources of animals, insect distribution, and collecting soil and water samples from habitats. Exploring the causes of this high complexity will improve our understanding of the ruminant gut microbiota and thus enable accurate modulation of it.

Interactions between animals and the environment or other individuals can lead to microbiota exchange, such as coliving, diet, lactation, etc (51–53). Most bacteria that enter a host are transient, whereas some bacteria can acquire a niche, allowing colonization (54), which may explain how the gut microbiome is shaped. The representative bacteria of *Verrucomicrobia* in the human gut is *Akkermansia*. Only two members (*A. glycaniphila* and *A. muciniphila*) and two *Candidatus* species (55) of the genus have been described, and only *A. muciniphila* has been found in the human gut. Phylogenetic trees showed that *Akkermansia* in the gut of Caprinae did not belong to any known *Akkermansia* clades of human gut origin, thus suggesting a high species-level diversity of *Akkermansia* in nonhuman hosts. Therefore, this implies that humans could acquire *Akkermansia* from animals, and only *A. muciniphila* has established a long-term symbiotic relationship with humans in the process of environmental adaptation. In addition, we also noticed *Treponema*, *Eggerthellaceae*, and *Euryarchaeota* special phylogenetic structures in the data set. To confirm this hypothesis, the challenge is to obtain more high-quality genomic information on animal-derived gut bacteria, to support a comparative genomic analysis and coevolution of related studies. A joint analysis of human and animal samples, as well as reading longer metagenomic sequencing, e.g., PacBio sequencing, will be useful (56, 57). In conclusion, this result provides an interesting insight into the origin of some members of the gut microbiota.

The geographical environment is an important factor affecting gut microbiota (58). The abundance of gut microbiota in Caprinae was largely changed, directly or indirectly, due to different geographical environments. Consistent with the phenomenon observed by Zhang et al. (14), there was no difference in the $\alpha$-diversity of samples from different regions. It is worth noting that the influences of diet, sex, age, species, and other factors are also the source of species structural heterogeneity (59, 60), and the complex relationship between these factors may be the reason why the specific species in different regions could not be proved. Tibet has very extreme living conditions, and the mammals living there face difficult survival challenges. The samples came from four species of Caprinae in Tibet that had adapted for living in the harsh highland environment. Adaptive evolution has led to benign changes in their gut microbiota in genes related to energy metabolism, helping them better adapt to the environment. Our results support the idea that highland Caprinae may be highly dependent on a specific gut microbiota for adaptation to the plateau environment. In

addition, this coevolving microbiota may also occur in other mammals, which requires additional samples to be included in future work.

By comparing the species communities of Tibetan and non-Tibetan samples, an accumulation of *Actinobacteria* was observed. Most of these actinomycetes were members of *Eggerthella* and *Olsenella*. The relative abundance of *Actinobacteria* was positively correlated with an elevation in a study of high-altitude pikas (61). Most of the actinomycetes were potential pathogens. *Eggerthella* has been associated with inflammatory diseases in humans, e.g., *Eggerthella lenta* is abundant in inflammatory bowel disease (IBD) patients (62). Although these bacteria have a potential disease risk, their niche in the plateau Caprinae remains unclear. In addition, samples from Tibetan areas contained many Ruminococcaceae members compared with those from other regions. These bacteria are involved in the synthesis of secondary bile acids and play an important role in gut metabolism (63). A study of highland yaks showed that the enterotype, characterized by *Ruminococcaceae*, was more stable and unaffected by seasonal changes (64). This suggests that enrichment of *Ruminococcaceae* plays a pivotal role in the adaptation of ruminants to high altitude environments. On the other hand, the gut of the Tibetan group was enriched with *Euryarchaeota*, most of which are methanogens. Functional enrichment analysis showed that the gut methane-related enzymes in Tibetan sheep were higher than that in non-Tibetan sheep. In fact, plateau ruminants are less capable of producing methane than ruminants at lower latitudes because of the energy loss associated with methane production (17, 65), which is not conducive to the efficient energy metabolism required to survive in harsh high-altitude environments. These plateau ruminants typically exhibit phenotypes with high volatile fatty acids, low methane production, and structural convergent variations in the gut microbiota (66). The methanogenic bacteria accumulated in the gut of Tibetan animals were probably some methylotrophic methanogens rather than *Methanobrevibacter gottschalkii*. Methylotrophic methanogens have been reported to be enriched in animals at high latitudes and associated with a low-methane phenotype (66–68). Together, these findings provide insights into changes in the structure and function of the gut microbiota of Caprinae in Tibet, which improves our understanding of how plateau ruminants adapt to their living environment.

Understanding the gut microbiota structure of Caprinae can help design appropriate interventions to improve feed conversion rates and individual health. This will require a deeper understanding of gut microbes in the future, such as substrate utilization, host interactions, and viral interactions. In addition, the high diversity of gut microbiota and the considerable unknown parts highlight the usefulness of developing simplified communities of cultured microorganisms that can be used to study molecular mechanisms in controlled experimental environments. In future work, many isolated and cultured species will be the basis for investigations of gut microbiota function. Deep metagenomic sequencing and species assembly of the gut microbiota are important to improve culture conditions and operate the gut microbiota.

## MATERIALS AND METHODS

**Experimental design and sample collection.** In this study, fecal samples from four Caprinae species (*Capra hircus*, $n = 18$; *Ovis aries*, $n = 9$; *Pantholops hodgsonii*, $n = 2$; and *Procapra picticaudata*, $n = 1$) from six provinces of China (Anhui, Jilin, Guangxi, Shandong, Shanxi, and Tibet) were collected (Table S1). The animals were healthy adults fed with plants and additives common in the area. To minimize pollution, the animals stood naturally during the sampling. One staff member calmed the animals and another staff member wore disposable sterile PE gloves and put his finger into the animal's rectum with distance of 5 cm to collect feces. For inaccessible animals (e.g., *P. hodgsonii*, also known as Tibetan antelopes), the method of tracking was adopted to collect the central part of fecal samples immediately after defecation; contact with the outside environment was avoided. Detailed information regarding animals was provided by local breeders or identified by professionals. The stool samples were then immediately transferred to sterile containers for homogenization and stored separately in DNase and RNase-free centrifuge tubes. All fresh stool samples were frozen in liquid nitrogen and mailed to the laboratory using dry ice within 24 h. Upon arrival, the samples were immediately stored in a −80℃ refrigerator for further experiments. The animal experiments were approved by Qingdao Agriculture University Ethics Committee.

**DNA extraction and sequencing.** Total bacterial DNA was extracted from each fecal sample (~200 mg per sample) using TIANGEN magnetic soil and stool DNA kit according to the manufacturer's instructions. DNA integrity was detected by 1% agarose gel electrophoresis, and genomic DNA concentration was measured by the Qubit DNA assay kit in a Qubit 3.0 fluorometer (Invitrogen, USA). A total amount of 0.2 $\mu$g DNA per sample was used as input material for the DNA library preparations. A metagenomic library with an insert size of 350 bp was generated using NEBNext Ultra DNA library prep kit for Illumina (NEB, USA) and index codes were added to each sample. Then DNA libraries were sequenced on the Illumina NovaSeq platform that generated 2 $\times$ 150 bp paired-end reads for a further analysis.

**Metagenomic assembly and binning.** All bioinformatic software and tools used in this study were set with default parameters unless otherwise specified. This study included 30 metagenomic samples representing a total of 3.34 Tb of data and 22.3 billion reads with a length of 150 bp. Illumina data were first cut by FASTP (v.0.19.5) (69) with options "-q 20 -u 30 -n 5 -y -Y 30 -l 90 –trim_poly_g." Low-quality bases at the ends of reads were cut, and too short and contaminated reads were filtered. To remove the contaminations from the host, high-quality reads controlled by quality control were mapped to the host genome derived from samples (the genome published by NCBI genome database) using Bowtie2 (v.2.4.1) (70), and the reads belonging to the host were excluded from the data. In the end, a total of 3.30 Tb of high-quality nonhost data were retained, averaging 110.1 Gb per sample, and these high-quality reads were performed for a subsequent analysis.

To obtain comprehensive genomic data, each sample was assembled using MEGAHIT (v.1.2.9) (71) with the optional parameter "–k-list 21,41,61,81,101,121,141." A total of 8,987,403 contigs with a minimum length of 500 bp were obtained, and the average $N_{50}$ length of these contigs was 10,319.9 bp. Reads were mapped to contigs using BWA MEM (v.0.7.17-R1188) (72) for producing SAM format files containing alignment information. SAMtools (v.0.1.19-44428CD) (73) was used to convert SAM files to BAM format. The sequencing depths of contigs were generated from the BAM files using the script *jgi_summarize_BAM_contig_depth* in the MetaBAT2 (v.2.12.1) (74) package. A total of 22,882 bins of contigs (>2 kb) were generated by using MetaBAT2 with options "-M 2000-s 200000 –saveCls –unbinned" based on sequence characteristics and sequencing depth of these contigs. Taxonomic classification of the bins was achieved using SpecI (v.1.0) (75) and GTDB-tk (v.1.1.1) (37). Within each sample, some bins were combined if similar sequencing depths ($\pm$10%), G+C content ($\pm$0.02), and identical species assignment were present. All bins were evaluated using the *lineage_wf* workflow of CheckM (v.1.1.2) (76), and only bins with integrity $\geq$70% and contamination $\leq$5% were retained for the subsequent analysis. dRep (v.2.5.4) (38) was used to remove redundancy with options "cluster drep -pa 0.99 -nc 0.3 – SkipSecondary" and all bins > 99% ANI were sorted into a collection. The quality score for each bin was calculated based on completeness $-5 \times$ contamination rate, and only the bins with the highest quality score in each collection were retained as MAGs. The detailed information of all MAGs was counted in Table S2, including the number of contigs, $N_{50}$ length, genome size, etc. High-quality nonhost reads were mapped back to MAGs using Bowtie2, and the relative abundances of MAGs in each sample were calculated accordingly.

**Species-level clustering of MAGs and comparison with other data sets.** To analyze species structure at the species level, dRep was used for further clustering of 5,046 MAGs. The MAGs with ANI > 95% were considered of the same species, and the MAGs with the highest quality score were screened out as representative species-level genome bins (SGBs) for each species, resulting in a set containing 3,306 SGBs. The ANI value between pairs of SGBs was calculated by FastANI (v.1.32) (26). A total of 21,315 high-quality MAGs were downloaded from four publicly available animal gut microbiota data sets (9, 11, 34, 35). All MAGs were used as reference genomes to identify new microbial genomes in our SGBs. Based on the species-level criterion (>95%ANI), these reference genomes were clustered with our SGBs using dRep, and the overlap between sets was mapped using UpSetR (v.1.4.0) (77).

**Taxonomic classification, phylogenetic, and functional analyses of MAGs.** Prokka (v.1.14.5) (78), a tool for rapid annotation of prokaryotic genomes, was employed to annotate all MAGs, including prediction of ORFs, rRNAs, and tRNAs. To determine the phylogenetic relationship of these MAGs, PhyloPhlAn (v.1.0) (79) was applied to build a phylogenetic tree of totaling 3,306 SGBs genomes by their protein subsequence. Taxonomic annotations were performed for all SGBs using the GTDB-tk (v.1.1.1) toolkit (37) based on Genome Taxonomy Database (GTDB). After the integration of the results, a manual correction was carried out to make the names conformable to the traditional nomenclature, and finally, the standardized classification labels in this study were generated.

All predicted proteins of MAGs were fully functionally characterized using the publicly available databases. The protein sequences of each MAG were blasted against the KEGG and CAZy databases using DIAMOND (80), and the functional terms were assigned according to the effective hit (E value < 1e-10, 50% protein coverage) in the databases. Further assignments of KEGG pathways and modules and CAZy families were provided by the KEGG (https://www.genome.jp/kegg/) and CAZy (http://www.cazy.org) websites, respectively. The number of genes encoding each function category (i.e., pathway, module, or CAZy family) was calculated by the total number of genes involved in the corresponding function. Then, according to the module composition provided by the KEGG website, the functional integrity of each module of MAGs was evaluated. The module with all structures was considered a complete module, the module missing one KOs was considered partial one, and the module missing two or more KOs was considered absent one (2, 1, and 0, respectively). Based on the EggNOG v.5.0 database (41), all MAGs were functionally annotated using eggNOG-mapper v2. The genes encoding microbial secondary metabolite BGC were predicted using antiSMASH (81). ABRicate (https://github.com/tseemann/abricate) was used to identify MAGs carrying antibiotic resistance genes, and the program

was searched based on the NCBI Bacterial Antibiotic Resistance Reference Gene Database, and CARD (82), ARG-ANNOT (83), and ResFinder (84) databases.

The differential enrichment KEGG modules were identified according to their reporter score (85, 86). One-tailed Wilcoxon rank-sum test was performed for all KOs presented in more than 3 samples, and multiple tests were adjusted using the Benjamin-Hochberg procedure to obtain Q values. Then, the z-score of each KO was calculated using the following formula: $Z_{KO_i} = \theta^{-1}(1-Q_{KO_i})$, where $\theta^{-1}$ is the inverse normal cumulative distribution and $Q_{KO_i}$ is the Q value for that KO. Finally, the reporter score ($Z_{module}$) of KEGG module was calculated by the following formula:

$$Z_{module} = \frac{1}{\sqrt{k}} \sum Z_{KO_i}$$

where $k$ is the number of KOs involved in the module. The report score ≥1.6 was used as the detection threshold of significantly differentiated modules.

**Phylogenetic of *Akkermansia, Treponema, Eggerthellales*, and *Euryarchaeota*.** PhyloPhlAn (v.1.0) was used to build the phylogenetic tree of these genera of concern. The reference genome sequences were from the NCBI database, and the GenBank ID was listed in Table S18. *Saccharomyces cerevisiae* YJM189 (GCA_000975735.4) reference genome was used as an outgroup for all phylogenetic trees. Interactive Tree Of Life (iTOL) v.6 (87) was used to visualize the phylogenetic tree. To clearly show the evolutionary relationships of species, the branches of the tree were proportionally deformed to fit the picture.

**Statistical analysis.** Statistical analyses were implemented at the R 4.0.3 platform. The map of China was drawn using maptools (v.1.1-1) and ggplot2 (v.3.3.5), and the SHP format data were obtained from Resource and Environment Science and Data Center. The rarefaction analysis was achieved through the *specaccum* function of the vegan (v.2.5-6) package. Principal coordinates analysis (PCoA) of Bray-Curtis distance was performed using the vegan package, which revealed the taxonomic or functional differences between pairs of samples or MAGs. The differences between groups were assessed using permutational multivariate analysis of variance (PERMANOVA) with 1,000 permutations, which was carried out with the *adonis* function of the vegan package. Also, alpha (Shannon and Simpson index) diversities were compared using the R vegan package, and the Wilcoxon rank-sum test was used to measure statistical differences in diversities among groups. Visualization of graphics was built in the R environment using the ggplot2, ComplexHeatmap (v.2.4.3), ggpubr (v.0.4.0), reshape2 (v.1.13.0), and ggtern (v.3.3.0) packages.

**Code availability.** The source codes related to graphic visualization and statistical analysis involved in this study is available on https://github.com/qb-lyu/caprinaeGut.

**Data availability.** The raw whole-metagenomic shotgun sequencing data acquired in this study had been deposited at the China Nucleotide Sequence Archive under the accession code PRJCA008889. All other data supporting the findings of this study are available in the paper and supplemental materials, or from the corresponding author(s) upon request. Other public data sets used in this study came from PRJEB33885 (34), PRJEB31266 (11), PRJEB35610 (35), and PRJNA657473 (9).

## SUPPLEMENTAL MATERIAL

Supplemental material is available online only.
**SUPPLEMENTAL FILE 1**, XLSX file, 1.4 MB.
**SUPPLEMENTAL FILE 2**, XLSX file, 0.6 MB.
**SUPPLEMENTAL FILE 3**, PDF file, 0.6 MB.

## ACKNOWLEDGMENTS

The study was supported by the National Funds for Supporting Reform and Development of Heilongjiang Provincial Colleges and Universities (grant no. 2022010009), the Longjiang Scholar Program of Heilongjiang Province (T201906), the Outstanding Youth Science Foundation of Heilongjiang province (JC2017007), and the Research Foundation for Distinguished Scholars of Qingdao Agricultural University (665-1120044, 665-1120046).

X.-X.Z., S.-H.L., D.-B.S., and H.-B.N. designed the study. S.-H.L., Q.-L.Y., and Q.-B.L. performed the research. S.-H.L. and Q.-B.L. analyzed data and wrote the paper. All authors contributed to the writing and revisions.

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
