## [Reviewer comments · Microbiology Spectrum]

Microbiology Spectrum

A catalog of over 5,000 metagenome-assembled microbial genomes from the Caprinae gut microbiota

Xiao-Xuan Zhang, Qing-Bo Lv, Qiulong Yan, Yue Zhang, Ruochun Guo, Jin-Xin Meng, He Ma, Si-Yuan Qin, Qinghe Zhu, Chunqiu Li, Rui Liu, Gang Liu, Shenghui Li, Dongbo Sun, and Hongbo Ni

Corresponding Author(s): Hongbo Ni, Heilongjiang Bayi Agricultural University

Review Timeline:

Submission Date:	June 13, 2022
Editorial Decision:	July 25, 2022
Revision Received:	September 24, 2022
Accepted:	October 10, 2022

Editor: Jennifer Auchtung

Reviewer(s): The reviewers have opted to remain anonymous.

Transaction Report:

DOI: <https://doi.org/10.1128/spectrum.02211-22>

July 25, 2022

Dr. Hongbo Ni
Heilongjiang Bayi Agricultural University
Daqing
China

Re: Spectrum02211-22 (A catalog of over 5,000 metagenome-assembled microbial genomes from the Caprinae gut microbiota)

Dear Dr. Hongbo Ni:

Thank you for submitting your manuscript to Microbiology Spectrum. As you will see in the reviews below, both reviewers made suggestions of changes that are needed to improve the quality of the manuscript. Please review and respond to all suggested changes, including the reviewers' suggestion to ensure the revised manuscript undergoes final editing for English grammar. When submitting the revised version of your paper, please provide (1) point-by-point responses to the issues raised by the reviewers as file type "Response to Reviewers," not in your cover letter, and (2) a PDF file that indicates the changes from the original submission (by highlighting or underlining the changes) as file type "Marked Up Manuscript - For Review Only". Please use this link to submit your revised manuscript - we strongly recommend that you submit your paper within the next 60 days or reach out to me. Detailed instructions on submitting your revised paper are below.

Link Not Available

Sincerely,

Jennifer Auchtung

Journals Department
Reviewer comments:

Reviewer #1 (Comments for the Author):

In this study, Zhang et al. examined the Caprinae fecal microbiome by constructing MAGs. However, the results are limited by sample size from 4 species. Moreover, the data analysis and writing need a greater improvement.

L4-6. Make a consistent style of author name.

L26. Gut is a very general definition, including ileum, cecum, colon and so on. Which region is focused on this study? and why?

L27. Based on the information in methods: fecal samples from four Caprinae species (*Capra hircus*, n = 18; *Ovis aries*, n = 9; *Pantholops hodgsonii*, n = 2; *Procavia picticaudata*, n = 1 from six provinces of China (Anhui, Jilin, Guangxi, Shandong, Shanxi, and Tibet)) were collected (Table S1).", the author only analyzed the fecal microbiome using shotgun sequencing. How to represent gut? Ruminant includes many species. Why this study only use goat and sheep animals? Only 1 sample were obtained from *Procavia picticaudata*. Recently, fecal metagenomes were also reported by many studies. It is appreciated if the

author can combine and use these data, although the author generated a deep sequence data for each sample. Then we can get a relatively comprehensive view of ruminant fecal microbiome.

L29-33. The author got MAGs, so what is the findings of these MAGs? But not generally describe the identification of MAGs. The enzymes and biosynthetic gene clusters are common results for metagenome study. What are the new findings and innovation?

L33. How the above results support co-evolution and feed efficiency? Make no sense.

L41-42. "Gut microbiota is considered to play a pivotal role in the energy conversion process of ruminants.". Rumen microbiome is much more important for energy harvest, although the lower gut microbiome is also critical for host healthy.

L43. How the GIT microbiome can enhance nutrient intake? List the evidence.

L47-54. Previous studies have reported the microbial gene catalog and constructed MAGs, so what is the significance of this study? It is accepted that rumen microbiome is complex than gut (usually represented by fecal samples) microbiome of monogastric animal. So how to conclude a high complexity of the gut microbiota of ruminants? What is unknown information based on the previous reports? Please give more supports for this hypothesis.

L58. References for "Caprinae animals have species polymorphism, including goats, sheep, antelopes, etc.".

L70-72. The environmental factors affected rumen and fecal microbiome. Why the author uses these ruminant species or breeds from only six provinces? Does these samples have clear representativeness, or can support the hypothesis in L58-69?

L86. The threshold of sequencing coverage and GC content and identical taxonomic assignment.

L97-98. Which version of GTDB? This affected the novelty of results.

L114-115. The author identified novel species using NCBI database. However, the method section indicated that "21,315 high-quality MAGs were downloaded from four publicly available animal gut microbiota datasets [9, 11, 33, 34], which represented the most comprehensive animal gut microbiota genome data available to date.". Why? Is there a representativeness for animal gut microbiome? The human and other mammals (pig) have generated a much greater database of MAGs.

L121-123. Why the genomes change?

L123-164. The author aimed to extend the phylogenetic tree of microorganisms using 4 databases based on the 3,306 SGBs. First, the author should include more MAGs database from other mammals, human and non-human primate. Second, the these SGBs were obtained from bins, but not culturable microorganisms, and its taxonomic classification is unclear. Thus, there maybe common MAGs form different database, and the author should combine these database together, and then analyzed the distribution. Third, this study generated MAGs from fecal samples, however, the database RUG were from rumen, and RGIG were from whole GIT of 7 ruminant species. The microbial profile must differ. The comparison makes no sense. Forth, please provide the value for the proportions. Fifth, be careful to use habitat and ecosystem.

L137. ruminant cattle?

L166-170. The author constructed the phylogenetic tree of Akkermansia. Based on the results in figure 2c, it seems that some MAGs showed very close phylogenetic relationship on nodes. So what is the node support? Moreover, a recent study has analyzed 112 Akkermansia muciniphila strains based on phylogenetic tree (10.1128/spectrum.02439-21). This indicate the database used in this study is in completed. Please defined "crossly presented".

L193-196. It is well known that the taxonomic classification absolutely affected the functional profile. The results for many studies have demonstrated the function of different phyla or families were distinct. So it is unreliable to get the conclusion "These differences in MAGs led to huge differences in the functional potential of the microbiota in the Caprinae gut". Comparing the results with fecal microbiome of other ruminants is warranted.

L204. Please define KEGG level B or C.

L208. Define a lot of.

L210. encoded Valine? Define a large numbers.

L259. gut MAGs.

L184-279. This study explored the functional profiles by describing and quantifying KEGG, CAZymes, BGCs and antibiotic-resistant genes. This is important, but did not give a deeper insight to these MAGs, particular to the host-specific and novel MAGs. A comparative genome analysis will be helpful.

L282. A significantly geographical effect was observed on bacterial community. So the bacterial community were based on MAGs or taxonomic classification?

L287-296. The results showed species, altitude, and sex were responsible for 16.1%, 8.8% and 8.7% of variations, respectively. Although a species must have a relative abundance of 0.01% in at least 50% of samples from a region to be considered universal in samples from that region. However, only 5 Tibet samples from 4 ruminant species, this will affect the comparison and may mask the real effect of hosts.

L294. Change region to geographical region.

L296. Change 260 to Two hundred and sixty.

L302-303. Again, all Tibet samples (n = 5) were compared with non-Tibet samples (n = 25). This comparison is difficult to reflect the real difference.

L317-318. Can bacteria synthesize bile acids?

L327-333. Repeat results, condense.

L334-336. The author only compared the different database. Because the size of database is different, thus the largely complemented and extended the existing reference can not reach. Combining these dataset together, and then analyze the taxonomic classification of raw reads, this can demonstrate the supplement to previous reference.

L337-338. The number of animals is more important than sequencing depth.

L342-344. Remove.

L345-346. A higher abundance of Ruminococcaceae in this study in comparison of gut core microbiota in other animals. How to

define core microbiota? And thus can conclude the complex of caprinae gut.

L348. What is other animals?

L349-350. Although the sheep are thought to be typically herbivores, their food sources in the wild may actually be more complex. Reference.

L357. What is "these bacteria"?

L358-359. "which may explain the source of some members and how the gut microbiome is shaped." What is some members?

L361. What is "The unique taxonomic characteristics"?

L363. "and this process occurs later than other gut bacteria". Make no sense.

L369-370. The species-level diversity of these bacteria may suggest that they have evolved longer in the animal gut than in humans. Make no sense.

L376-383. Meaningless. Discuss the novel findings, such as the changed genome size of MAGs from different phyla.

L386-388. The diversity indices were not significantly changed, so how can conclude "This suggests that the geographic heterogeneity of gut microbiota may be caused by some species that metabolize specific substrates."?

L392-397. Provide evidences.

L449. Which kit? The DNA extraction methods have a clear effect on sequencing quality.

L464. Is there a Procpra picticaudata genome in NCBI database?

Reviewer #2 (Comments for the Author):

Comments:

In this manuscript, the authors constructed metagenome-assembled reference genome catalog from Caprinae gut microbiota using ultra-deep metagenomics sequencing. This topic is extreme important nowadays because it is sooner and easier to acquire high quality genome of uncultured bacteria than traditional isolation and sequencing pipeline. Overall, the paper is well written (although the English could be improved, it was not a significant impediment to understanding), the experiments well designed, and the statistical analysis relevant and robust.

I have only minor suggestions for improvement/clarification.

1. Line 94-95: the author clustered the MAG to establish SGB using 95% ANI. However, in material and methods, Line 481-482, 99% ANI was used in dRep to perform dereplication which was normally used to establish strain level cluster of MAG. Please clarify this.

2. Line 468-469: single sample assembly strategy was employed in this manuscript to acquire contig catalog. For metagenomics sequencing, co-assembly is also widely used to further refine and improve the assembly output of single sample assembly strategy, especially for ultra-deep metagenome sequencing. And this may have considerable effect on downstream binning output. Please clarify this issue.

Staff Comments:

Preparing Revision Guidelines

Please return the manuscript within 60 days; if you cannot complete the modification within this time period, please contact me. If you do not wish to modify the manuscript and prefer to submit it to another journal, please notify me of your decision immediately so that the manuscript may be formally withdrawn from consideration by Microbiology Spectrum.

If your manuscript is accepted for publication, you will be contacted separately about payment when the proofs are issued;

please follow the instructions in that e-mail. Arrangements for payment must be made before your article is published. For a complete list of **Publication Fees**, including supplemental material costs, please visit our website.

In this study, Zhang et al. examined the Caprinae fecal microbiome by constructing MAGs. However, the results are limited by sample size from 4 species. Moreover, the data analysis and writing need a greater improvement.

L4-6. Make a consistent style of author name.

L26. Gut is a very general definition, including ileum, cecum, colon and so on. Which region is focused on this study? and why?

L27. Based on the information in methods: fecal samples from four Caprinae species (*Capra hircus*, n = 18; *Ovis aries*, n = 9; *Pantholops hodgsonii*, n = 2; *Procapra picticaudata*, n = 1 from six provinces of China (Anhui, Jilin, Guangxi, Shandong, Shanxi, and Tibet)) were collected (Table S1).”, the author only analyzed the fecal microbiome using shotgun sequencing. How to represent gut? Ruminant includes many species. Why this study only use goat and sheep animals? Only 1 sample were obtained from *Procapra picticaudata*. Recently, fecal metagenomes were also reported by many studies. It is appreciated if the author can combine and use these data, although the author generated a deep sequence data for each sample. Then we can get a relatively comprehensive view of ruminant fecal microbiome.

L29-33. The author got MAGs, so what is the findings of these MAGs? But not generally describe the identification of MAGs. The enzymes and biosynthetic gene clusters are common results for metagenome study. What are the new findings and innovation?

L33. How the above results support co-evolution and feed efficiency? Make no sense.

L41-42. “Gut microbiota is considered to play a pivotal role in the energy conversion process of ruminants.”. Rumen microbiome is much more important for energy harvest, although the lower gut microbiome is also critical for host healthy.

L43. How the GIT microbiome can enhance nutrient intake? List the evidence.

L47-54. Previous studies have reported the microbial gene catalog and constructed MAGs, so what is the significance of this study? It is accepted that rumen microbiome is complex than gut (usually represented by fecal samples) microbiome of monogastric animal. So how to conclude a high complexity of the gut microbiota of ruminants? What is unknown information based on the previous reports? Please give more supports for this hypothesis.

L58. References for “Caprinae animals have species polymorphism, including goats, sheep, antelopes, etc.”.

L70-72. The environmental factors affected rumen and fecal microbiome. Why the author uses these ruminant species or breeds from only six provinces? Does these samples have clear representativeness, or can support the hypothesis in L58-69?

L86. The threshold of sequencing coverage and GC content and identical taxonomic assignment.

L97-98. Which version of GTDB? This affected the novelty of results.

L114-115. The author identified novel species using NCBI database. However, the method section indicated that “21,315 high-quality MAGs were downloaded from four publicly available animal gut microbiota datasets [9, 11, 33, 34], which represented the most comprehensive animal gut microbiota genome data available to date.”. Why? Is there a representativeness for animal gut microbiome? The human and other mammals (pig) have generated a much greater database of MAGs.

L121-123. Why the genomes change?

L123-164. The author aimed to extend the phylogenetic tree of microorganisms using 4

databases based on the 3,306 SGBs. First, the author should include more MAGs database from other mammals, human and non-human primate. Second, these SGBs were obtained from bins, but not culturable microorganisms, and its taxonomic classification is unclear. Thus, there may be common MAGs from different databases, and the author should combine these databases together, and then analyze the distribution. Third, this study generated MAGs from fecal samples, however, the database RUG were from rumen, and RGIG were from whole GIT of 7 ruminant species. The microbial profile must differ. The comparison makes no sense. Fourth, please provide the value for the proportions. Fifth, be careful to use habitat and ecosystem.

L137. ruminant cattle?

L166-170. The author constructed the phylogenetic tree of Akkermansia. Based on the results in figure 2c, it seems that some MAGs showed very close phylogenetic relationship on nodes. So what is the node support? Moreover, a recent study has analyzed 112 Akkermansia muciniphila strains based on phylogenetic tree (10.1128/spectrum.02439-21). This indicates the database used in this study is incomplete. Please define "crossly presented".

L193-196. It is well known that the taxonomic classification absolutely affected the functional profile. The results for many studies have demonstrated the function of different phyla or families were distinct. So it is unreliable to get the conclusion "These differences in MAGs led to huge differences in the functional potential of the microbiota in the Caprinae gut". Comparing the results with fecal microbiome of other ruminants is warranted.

L204. Please define KEGG level B or C.

L208. Define a lot of.

L210. encoded Valine? Define a large numbers.

L259. gut MAGs.

L184-279. This study explored the functional profiles by describing and quantifying KEGG, CAZymes, BGCs and antibiotic-resistant genes. This is important, but did not give a deeper insight to these MAGs, particularly to the host-specific and novel MAGs. A comparative genome analysis will be helpful.

L282. A significantly geographical effect was observed on bacterial community. So the bacterial community were based on MAGs or taxonomic classification?

L287-296. The results showed species, altitude, and sex were responsible for 16.1%, 8.8% and 8.7% of variations, respectively. Although a species must have a relative abundance of 0.01% in at least 50% of samples from a region to be considered universal in samples from that region. However, only 5 Tibet samples from 4 ruminant species, this will affect the comparison and may mask the real effect of hosts.

L294. Change region to geographical region.

L296. Change 260 to Two hundred and sixty.

L302-303. Again, all Tibet samples (n = 5) were compared with non-Tibet samples (n = 25). This comparison is difficult to reflect the real difference.

L317-318. Can bacteria synthesize bile acids?

L327-333. Repeat results, condense.

L334-336. The author only compared the different database. Because the size of database is different, thus the largely complemented and extended the existing reference can not reach. Combining these datasets together, and then analyze the taxonomic classification of raw reads,

this can demonstrate the supplement to previous reference.

L337-338. The number of animals is more important than sequencing depth.

L342-344. Remove.

L345-346. A higher abundance of Ruminococcaceae in this study in comparison of gut core microbiota in other animals. How to define core microbiota? And thus can conclude the complex of caprinae gut.

L348. What is other animals?

L349-350. Although the sheep are thought to be typically herbivores, their food sources in the wild may actually be more complex. Reference.

L357. What is "these bacteria"?

L358-359. "which may explain the source of some members and how the gut microbiome is shaped." What is some members?

L361. What is "The unique taxonomic characteristics"?

L363. "and this process occurs later than other gut bacteria". Make no sense.

L369-370. The species-level diversity of these bacteria may suggest that they have evolved longer in the animal gut than in humans. Make no sense.

L376-383. Meaningless. Discuss the novel findings, such as the changed genome size of MAGs from different phyla.

L386-388. The diversity indices were not significantly changed, so how can conclude "This suggests that the geographic heterogeneity of gut microbiota may be caused by some species that metabolize specific substrates."?

L392-397. Provide evidences.

L449. Which kit? The DNA extraction methods have a clear effect on sequencing quality.

L464. Is there a *Procopra picticaudata* genome in NCBI database?

20 September, 2022

Prof. Jennifer Auchtung
Editor-in-Chief
Microbiology Spectrum

Dear Prof. Jennifer Auchtung,

Re: Revised Manuscript ID Spectrum02211-22

On behalf of all co-authors, I would like to thank you and the two reviewers very much for their careful review and constructive suggestions with regard to our manuscript (MS) ID **Spectrum02211-22**. These comments are all valuable and very helpful for revising and improving our paper, as well as the importance guiding significance to our MS. We have studied comments carefully and have made correction which we hope meet with approval.

Responses to the comments and suggestions of Reviewer #1:

Reviewer #1 (Comments for the Author):

In this study, Zhang et al. examined the Caprinae fecal microbiome by constructing MAGs. However, the results are limited by sample size from 4 species. Moreover, the data analysis and writing need a greater improvement.

Point 1. L4-6. Make a consistent style of author name.

Responses: We thank **Reviewer #1** very much for his/her constructive comments and suggestions on our MS. We have revised our manuscript according to the comments.

Point 2. L26. Gut is a very general definition, including ileum, cecum, colon and so on. Which region is focused on this study? and why?

Responses: We thank **Reviewer #1** very much for his/her constructive comments and suggestions on our MS. Our study focused on the fecal microbiota, which is also commonly referred to as the gut microbiota (**Chen C et al. 2021 DOI: 10.1038/s41467-021-21295-0; Junjie Qin et al. 2012 DOI: 10.1038/nature11450**). Ruminant microbiota is important, but gut microbiota plays an indispensable role. In recent years, a large number of rumen microbiota datasets have been published, while gut microbiota datasets still need to be refined, especially those related to sheep. Therefore, we used deep metagenomic sequencing to have an understanding of the intestinal flora of sheep and obtain their genomes to support future research.

Point 3. L27. Based on the information in methods: fecal samples from four Caprinae species (*Capra hircus*, n = 18; *Ovis aries*, n = 9; *Pantholops hodgsonii*, n = 2; *Procapra picticaudata*, n = 1 from six provinces of China (Anhui, Jilin, Guangxi, Shandong, Shanxi, and Tibet)) were collected (Table S1).”, the author only analyzed the fecal microbiome using shotgun sequencing. How to represent gut? Ruminant includes many species. Why this study only uses goat and sheep animals? Only 1 sample were obtained from *Procapra picticaudata*. Recently, fecal metagenomes were also reported by many studies. It is appreciated if the author can combine and use these data, although the author generated a deep sequence data for each sample. Then we can get a relatively comprehensive view of ruminant fecal microbiome.

Responses: We thank **Reviewer #1** very much for his/her constructive comments and suggestions on our MS. We only collected feces from these animals in order to extend our understanding of the extant gut microbiota of Caprinae species through metagenomic sequencing. We acknowledge that these data are not comprehensive and there is no way to represent the whole gut, but as a supplement, we think they are of some interest. In addition, the aim of our study was to expansion the genomes of bacteria and archaea in the gut microbiota of the Caprinae species. Therefore, other ruminants were not considered in the current study.

Point 4. L29-33. The author got MAGs, so what is the findings of these MAGs? But not generally describe the identification of MAGs. The enzymes and biosynthetic gene clusters are common results for metagenome study. What are the new findings and innovation?

Responses: We thank **Reviewer #1** very much for his/her constructive comments and suggestions on our MS. A large proportion of our assembled MAGs are previously undiscovered new species, extending the current catalog of gut microbes at the genomic level. As supplementary data on gut microbiota community resources of Caprinae species, we believe that these data are innovative to some extent.

Point 5. L33. How the above results support co-evolution and feed efficiency? Make no sense.

Responses: We thank **Reviewer #1** very much for his/her constructive comments and suggestions on our MS. The expectation of this study was to evaluate the role of commensal bacteria in coevolution, but these results were not ideal. Thanks for your instructive advice, we have removed the argument.

Point 6. L41-42. “Gut microbiota is considered to play a pivotal role in the energy conversion process of ruminants.”. Rumen microbiome is much more important for energy harvest, although the lower gut microbiome is also critical for host healthy.

Responses: We thank **Reviewer #1** very much for his/her constructive comments and suggestions on our MS. Thank you for your advice, here is a misstatement. We have changed the description here in the manuscript.

Point 7. L43. How the GIT microbiome can enhance nutrient intake? List the evidence.

Responses: We thank **Reviewer #1** very much for his/her constructive comments and suggestions on our MS. The statement here is ambiguous and we have changed it to make it clearer. As far as enhancement of nutrient intake, GIT microbiome can provide essential capacities for the fermentation of nondigestible substrates like dietary fibers and endogenous intestinal mucus, produce short chain fatty acids and gases, strengthen metabolism of phytochemicals and xenobiotics and et al, as reviewed previously.

Point 8. L47-54. Previous studies have reported the microbial gene catalog and constructed MAGs, so what is the significance of this study? It is accepted that rumen microbiome is complex than gut (usually represented by fecal samples) microbiome of monogastric animal. So how to conclude a high complexity of the gut microbiota of ruminants? What is unknown information based on the previous reports? Please give more supports for this hypothesis.

Responses: We thank **Reviewer #1** very much for his/her constructive comments and suggestions on our MS. The main significance of this study is to expand the existing genomic resources of the gut microbiota and improve the understanding of the gut microbiota environment of Caprinae species. With respect to gastrointestinal microbiome complexity, the expression in the manuscript is not rigorous enough. So, we change "gut microbiota" to "gastrointestinal microbiome". "Unknow information" means that the gut microbiota structure characteristics of different species of Caprinae in different regions have not been fully recognized.

Point 9. L58. References for “Caprinae animals have species polymorphism, including goats, sheep, antelopes, etc.”.

Responses: We thank **Reviewer #1** very much for his/her constructive comments and suggestions on our MS. We have cited the relevant reference. Up until now, the subfamily Caprinae has been included 15 genera and 74 species, supporting view of species polymorphism.

Point 10. L70-72. The environmental factors affected rumen and fecal microbiome. Why the author uses these ruminant species or breeds from only six provinces? Does these samples have clear representativeness, or can support the hypothesis in L58-69?

Responses: We thank **Reviewer #1** very much for his/her constructive comments and suggestions on our MS. We collected samples from six different regions in China, far apart from each other. In addition, these animals live in the wild with little or no contact with humans, and they are representative of the region to some extent. PCoA analysis showed that the samples from the same region were close to each other, which suggested that these samples had some regional consistency.

Point 11. L86. The threshold of sequencing coverage and GC content and identical taxonomic assignment.

Responses: We thank **Reviewer #1** very much for his/her constructive comments and suggestions on our MS. The sequencing coverage was $\pm 10\%$, the GC content was $\pm 2\%$, and the species analysis was consistent with MAG merging.

Point 12. L97-98. Which version of GTDB? This affected the novelty of results.

Responses: We thank **Reviewer #1** very much for his/her constructive comments and suggestions on our MS. We have revised our manuscript according to the comments. We used GTDB R89 database, release date: June 17, 2019.

Point 13. L114-115. The author identified novel species using NCBI database. However, the method section indicated that “21,315 high-quality MAGs were downloaded from four publicly available animal gut microbiota datasets [9, 11, 33, 34], which represented the most comprehensive animal gut microbiota genome data available to date.”. Why? Is there a representativeness for animal gut microbiome? The human and other mammals (pig) have generated a much greater database of MAGs.

Responses: We thank **Reviewer #1** very much for his/her constructive comments and suggestions on our MS. We selected the most representative ruminant gastrointestinal microbiota datasets, nonhuman primate and human databases. We acknowledge that these databases are not comprehensive, so we have removed this sentence.

Point 14. L121-123. Why the genomes change?

Responses: We thank **Reviewer #1** very much for his/her constructive comments and suggestions on our MS. The classification of GTDB is determined by the similarity of marker genes. Based on this rule, genome size seems to have little effect. Moreover, the smallest actinomycetes genome in NCBI was 0.00155Mbp with a GC content of 34.12%, while the largest genome was 14.33 Mbp with a G+C content of 77.01%. Furthermore, genomic features vary considerably and may actually be due to gene copying and loss (**Ventura M et al. 2007 DOI:10.1128/MMBR.00005-07**).

Point 15. L123-164. The author aimed to extend the phylogenetic tree of microorganisms using 4 databases based on the 3,306 SGBs. First, the author should include more MAGs database from other mammals, human and non-human primate. Second, the these SGBs were obtained from bins, but not culturable microorganisms, and its taxonomic classification is unclear. Thus, there maybe common MAGs form different database, and the author should combine these database together, and then analyzed the distribution. Third, this study generated MAGs

from fecal samples, however, the database RUG were from rumen, and RGIG were from whole GIT of 7 ruminant species. The microbial profile must differ. The comparison makes no sense. Forth, please provide the value for the proportions. Fifth, be careful to use habitat and ecosystem.

Responses: We thank **Reviewer #1** very much for his/her constructive comments and suggestions on our MS. Although we did not include the new mammalian data set, the current data set is still accepted. We refer to the research of Xie et al. (doi: 10.1186/s40168-021-01078-x), and include the dataset in this study, which is representative to some extent. Most SGBs did not overlap with these datasets, indicating that they are potentially novel bacteria. We will further isolate these new bacteria in subsequent studies. We acknowledge that comparing microbial differences across profiles is of little significance, but our aim is to augment these databases.

Point 16. L137. ruminant cattle?

Responses: We thank **Reviewer #1** very much for his/her constructive comments and suggestions on our MS. We have revised our manuscript according to the comments.

Point 17. L166-170. The author constructed the phylogenetic tree of *Akkermansia*. Based on the results in figure 2c, it seems that some MAGs showed very close phylogenetic relationship on nodes. So what is the node support? Moreover, a recent study has analyzed 112 *Akkermansia muciniphila* strains based on phylogenetic tree (10.1128/spectrum.02439-21). This indicate the database used in this study is in completed. Please defined “crossly presented”.

Responses: We thank **Reviewer #1** very much for his/her constructive comments and suggestions on our MS. The evolutionary analysis revealed a long evolutionary distance between the sheep derived *Akkermansia* and the four major phylogroups of *Akkermansia* isolated from the human gut, implying different evolutionary selection pressures. "crossly presented" was an error in the presentation, and we have modified the statement.

Point 18. L193-196. It is well known that the taxonomic classification absolutely affected the functional profile. The results for many studies have demonstrated the function of different phyla or families were distinct. So it is unreliable to get the conclusion “These differences in MAGs led to huge differences in the functional potential of the microbiota in the Caprinae gut”. Comparing the results with fecal microbiome of other ruminants is warranted.

Responses: We thank **Reviewer #1** very much for his/her constructive comments and suggestions on our MS. Sorry, our conclusion here is not rigorous, has been deleted.

Point 19. L204. Please define KEGG level B or C. L208. Define a lot of.

Responses: We thank **Reviewer #1** very much for his/her constructive comments and suggestions on our MS. KO is defined as level D, ascending in order according to the hierarchy.

Point 20. L210. encoded Valine? Define a large numbers.

Responses: We thank **Reviewer #1** very much for his/her constructive comments and suggestions on our MS. Compared to other portal actinomycetes in valine expression, the data is shown in Table S9.

Point 21. L259. gut MAGs.

Responses: We thank **Reviewer #1** very much for his/her constructive comments and suggestions on our MS. We have revised our manuscript according to the comments.

Point 22. L184-279. This study explored the functional profiles by describing and quantifying KEGG, CAZymes, BGCs and antibiotic-resistant genes. This is important, but did not give a deeper insight to these MAGs, particular to the host-specific and novel MAGs. A comparative genome analysis will be helpful.

Responses: We thank **Reviewer #1** very much for his/her constructive comments and suggestions on our MS. Due to sampling limitations, we were unable to analyze aspects such as host specificity, so only the functional maps of these MAGs were described in this study. In the subsequent work, we will further improve these analyses and experiments.

Point 23. L282. A significantly geographical effect was observed on bacterial community. So the bacterial community were based on MAGs or taxonomic classification?

Responses: We thank **Reviewer #1** very much for his/her constructive comments and suggestions on our MS. Based on abundance of MAGs.

Point 24. L287-296. The results showed species, altitude, and sex were responsible for 16.1%, 8.8% and 8.7% of variations, respectively. Although a species must have a relative abundance of 0.01% in at least 50% of samples from a region to be considered universal in samples from that region. However, only 5 Tibet samples from 4 ruminant species, this will affect the comparison and may mask the real effect of hosts.

Responses: We thank **Reviewer #1** very much for his/her constructive comments and suggestions on our MS. True results may obscure true information because of the small number of samples. But we did find through our analysis that the flora of the animals in the highlands contained flora associated with increased oxygen use. Subsequent studies will expand the sample size to test this idea separately.

Point 25. L294. Change region to geographical region.

Responses: We thank **Reviewer #1** very much for his/her constructive comments and suggestions on our MS. We have revised our manuscript according to the comments.

Point 26. L296. Change 260 to Two hundred and sixty.

Responses: We thank **Reviewer #1** very much for his/her constructive comments and suggestions on our MS. We have revised our manuscript according to the comments.

Point 27. L302-303. Again, all Tibet samples (n = 5) were compared with non-Tibet samples (n = 25). This comparison is difficult to reflect the real difference.

Responses: We thank **Reviewer #1** very much for his/her constructive comments and suggestions on our MS. The smaller sample size does affect the credibility of the results, but it is still statistically plausible. We will supplement these contents in the subsequent study.

Point 28. L317-318. Can bacteria synthesize bile acids?

Responses: We thank **Reviewer #1** very much for his/her constructive comments and suggestions on our MS. Functional annotation revealed the presence of genes involved in bile acid synthesis in these bacteria, and we modified the description here to reduce ambiguity.

Point 29. L327-333. Repeat results, condense.

Responses: We thank **Reviewer #1** very much for his/her constructive comments and suggestions on our MS. We removed the superfluous description.

Point 30. L334-336. The author only compared the different database. Because the size of database is different, thus the largely complemented and extended the existing reference can not reach. Combining these dataset together, and then analyze the taxonomic classification of raw reads, this can demonstrate the supplement to previous reference.

Responses: We thank **Reviewer #1** very much for his/her constructive comments and suggestions on our MS. Due to resource issues, these analyses could not be supplemented at a short time. But the results are still relevant.

Point 31. L337-338. The number of animals is more important than sequencing depth.

Responses: We thank **Reviewer #1** very much for his/her constructive comments and suggestions on our MS. We agree with you that increasing sequencing depth does not in fact improve the results much.

Point 32. L342-344. Remove.

Responses: We thank **Reviewer #1** very much for his/her constructive comments and suggestions on our MS. We

have deleted this sentence.

Point 33. L345-346. A higher abundance of Ruminococcaceae in this study in comparison of gut core microbiota in other animals. How to define core microbiota? And thus can conclude the complex of caprinae gut.

Responses: We thank **Reviewer #1** very much for his/her constructive comments and suggestions on our MS. Our screening threshold was that SGB with 1% coverage in 90% of the samples were considered as core species.

Point 34. L348. What is other animals?

Responses: We thank **Reviewer #1** very much for his/her constructive comments and suggestions on our MS. This refers to hosts of other databases, including ruminants, nonhuman primates, and humans.

Point 35. L349-350. Although the sheep are thought to be typically herbivores, their food sources in the wild may actually be more complex. Reference.

Responses: We thank **Reviewer #1** very much for his/her constructive comments and suggestions on our MS. This is just a speculation, and we can't find a usable reference. We have modified the expression here.

Point 36. L357. What is “these bacteria”?

Responses: We thank **Reviewer #1** very much for his/her constructive comments and suggestions on our MS. We have changed the description here. These bacteria are bacteria that enter the host through microbiota exchange.

Point 37. L358-359. “which may explain the source of some members and how the gut microbiome is shaped.”
What is some members?

Responses: We thank **Reviewer #1** very much for his/her constructive comments and suggestions on our MS. We changed the statement here.

Point 38. L361. What is “The unique taxonomic characteristics”?

Responses: We thank **Reviewer #1** very much for his/her constructive comments and suggestions on our MS. We deleted the sentence

Point 39. L363. “and this process occurs later than other gut bacteria”. Make no sense.

Responses: We thank **Reviewer #1** very much for his/her constructive comments and suggestions on our MS. We deleted the sentence

Point 40. L369-370. The species-level diversity of these bacteria may suggest that they have evolved longer in the animal gut than in humans. Make no sense.

Responses: We thank **Reviewer #1** very much for his/her constructive comments and suggestions on our MS. We deleted the sentence

Point 41. L376-383. Meaningless. Discuss the novel findings, such as the changed genome size of MAGs from different phyla.

Responses: We thank **Reviewer #1** very much for his/her constructive comments and suggestions on our MS. We deleted the discussion.

Point 42. L386-388. The diversity indices were not significantly changed, so how can conclude “This suggests that the geographic heterogeneity of gut microbiota may be caused by some species that metabolize specific substrates.”?

Responses: We thank **Reviewer #1** very much for his/her constructive comments and suggestions on our MS. We deleted the discussion.

Point 43. L392-397. Provide evidences.

Responses: We thank **Reviewer #1** very much for his/her constructive comments and suggestions on our MS. We have made inferences based on the present results of this study, but there is not enough evidence to support these inferences. We will further prove this in subsequent studies.

Point 44. L449. Which kit? The DNA extraction methods have a clear effect on sequencing quality.

Responses: We thank **Reviewer #1** very much for his/her constructive comments and suggestions on our MS. We add the type and brand of DNA extraction kit.

Point 45. L464. Is there a *Procapra picticaudata* genome in NCBI database?

Responses: We thank **Reviewer #1** very much for his/her constructive comments and suggestions on our MS. NCBI does not have a genome for *Procapra picticaudata*, and we used sheep and goat genomes as reference genomes.

Responses to the comments and suggestions of Reviewer #2:

Reviewer #2 (Comments for the Author):

In this manuscript, the authors constructed metagenome-assembled reference genome catalog from Caprinae gut microbiota using ultra-deep metagenomics sequencing. This topic is extremely important nowadays because it is sooner and easier to acquire high quality genome of uncultured bacteria than traditional isolation and sequencing pipeline. Overall, the paper is well written (although the English could be improved, it was not a significant impediment to understanding), the experiments well designed, and the statistical analysis relevant and robust.

I have only minor suggestions for improvement/clarification.

Point 1. Line 94-95: the author clustered the MAG to establish SGB using 95% ANI. However, in material and methods, Line 481-482, 99% ANI was used in dRep to perform dereplication which was normally used to establish strain level cluster of MAG. Please clarify this.

Responses: We thank **Reviewer #2** very much for his/her constructive comments and suggestions on our MS. The clustering strategy described in lines 481-482 is a de-redundancy operation for MAGs, and in lines 492-493 the SGB is clustered with a threshold of 95% ANI.

Point 2. Line 468-469: single sample assembly strategy was employed in this manuscript to acquire contig catalog. For metagenomics sequencing, co-assembly is also widely used to further refine and improve the assembly output of single sample assembly strategy, especially for ultra-deep metagenome sequencing. And this may have considerable effect on downstream binning output. Please clarify this issue.

Responses: We thank **Reviewer #2** very much for his/her constructive comments and suggestions on our MS. Due to resource limitation, co-assembly was not performed in this study.

October 10, 2022

Dr. Hongbo Ni
Heilongjiang Bayi Agricultural University
Daqing
China

Re: Spectrum02211-22R1 (A catalog of over 5,000 metagenome-assembled microbial genomes from the Caprinae gut microbiota)

Dear Dr. Hongbo Ni:

Your manuscript has been accepted, and I am forwarding it to the ASM Journals Department for publication. You will be notified when your proofs are ready to be viewed.

Sincerely,

Jennifer Auchtung
Editor, Microbiology Spectrum

Journals Department
Supplemental Material: Accept
Supplemental Material: Accept
Supplemental Material: Accept